# Digital Image Correlation Characterization of Deformation Behavior and Cracking of Porous Segmented Alumina under Uniaxial Compression

Vladimir Kibitkin [1,*], Nickolai Savchenko [1,*], Mikhail Grigoriev [2], Andrey Solodushkin [1], Alexander Burlachenko [1], Ales Buyakov [1], Anna Zykova [1], Valery Rubtsov [1] and Sergei Tarasov [1,*]

[1] Institute of Strength Physics and Materials Science, Siberian Branch of Russian Academy of Sciences, 634055 Tomsk, Russia
[2] Laboratory of Nanotechnologies of Metallurgy, National Research Tomsk State University, 634050 Tomsk, Russia

* Correspondence: vvk@ispms.ru (V.K.); savnick@ispms.ru (N.S.); tsy@ispms.ru (S.T.);
Tel.: +7-906-955-45-41 (N.S.); +7-923-420-10-14 (S.T.)

**Abstract:** In this study, the two-dimensional digital image correlation (DIC) technique has been applied to sequences of images taken from the surfaces of porous, segmented alumina samples during uniaxial compression tests. The sintered alumina was structurally composed of polycrystalline alumina grains with interior ~3–5-μm pores, a network of discontinuities that subdivided the sample into ~230 μm segments, and ~110 μm pores located at the discontinuity network nodes. Bimodal pore structure and the segment boundaries were the results of the evaporation and the outgassing of the paraffin and ultra-high-molecular-weight polyethylene admixed with alumina powder via slip casting. Only partial bonding bridges between the segments were formed during a low-temperature sintering at 1300 °C for 1 h. A special technological approach made it possible to change the strength of the partial bonding bridges between the segments, which significantly affected the deformation behavior ceramics during compression. The subpixel accuracy of the DIC results was achieved using an interpolation scheme for the identification functional. The vector fields obtained in the experiment made it possible to characterize the processes of deformation and destruction of a porous, segmented alumina using the strain localization in situ maps, cardinal plastic shear, and circulation of vector fields. The use of these characteristics made it possible to reveal new details in the mechanisms of deformation and destruction of segmented ceramics. The localizations of damage were identified and related to the characteristic structural heterogeneities of the tested porous segmented ceramics.

**Keywords:** segmented ceramic; digital image correlation





## 1. Introduction

Ceramic materials are usually considered as brittle solids with only elastic behavior under loading, which is followed by instability fracture stage. However, some of them can be described as quasi-brittle ones when revealing their non-linear behavior under loading [1–5]. Depending upon when this non-linear behavior is observed, i.e., either before or after achieving the maximum load, all ceramic materials may be classified as damageable or strain-softening ceramics [6–8].

The quasi-brittle behavior of ceramics is a function of many structural toughening mechanisms that can be classified as either intrinsic or extrinsic [6]. The intrinsic mechanisms act in front of a crack tip and independently of the crack size or geometry in ceramic materials made of crystals with anisotropic thermal expansion coefficients [9,10] or containing zirconia capable of transformation toughening [11], as well as natural rocks with micro-crack propagation, etc. [12].

The extrinsic structural toughening mechanisms relate to sliding or interlocking between two rough fracture surfaces formed in multi-layer or segmented structures [6–8,13–16]. These materials possess complex hierarchical structures composed of either fully dense or porous ceramic segments with only weak bonding at the segment boundaries.

These segments can either be bonded together by means of a binder or form a relatively rigid macrostructure of geometrically interlocked segments without any chemical bonding [13]. A crack nucleated in a single segment can propagate along the boundaries filled with the binder and thus easily lead to brittle fracture by propagation of the main crack. The geometrically interlocked segments do not offer such an intersegment crack propagation because any incipient crack would come to rest against the next segment. It was shown [6,8,13,14] that segmented structure allows, after reaching the ultimate strength, to maintain a constant level of stress at a relative strain significantly higher than that of the monolithic ceramics, due to the fact that individual blocks of the material may undergo local displacements and rotations relative to their neighbors under constrained compression conditions [6,8,13,14].

When such a segmented ceramic sample is quasi-statically loaded by compression force, it is capable of deforming by means of the segment relative sliding and rotation without trans-segment deformation or fracture [6]. Given that both siding and rotation are accompanied by friction, such a segmented ceramic material can be capable of dissipating much more mechanical energy as compared to that of a monolithic sample [6].

Intrinsic, as well as extrinsic, structural toughening mechanisms have been investigated by the example of ceramic materials with weakly bonded interfaces [6]. A complex method that allowed combination of a numerical model with three-dimensional digital image correlation (DIC) has been developed for creating a bio-inspired multi-layer architecture of ceramics and studying their toughening mechanisms under low-velocity impact loading [6]. Both extrinsic and intrinsic toughening mechanisms were captured: sliding of the tiles in the architectured ceramics and channel plastic deformation in adhesive interlayers, respectively. It was shown [6] that a fine balance of the interlocking and architectured ceramic panels block sizes serves for controlled frictional sliding and rotation of blocks, minimizes damage of the individual blocks, and optimizes performance of the whole structure.

Alumina–mullite porous ceramic plates were segmented into topologically interconnected blocks with a special osteomorphic geometry [8]. In addition to high sound absorption, a combination of mechanical properties of segmented plates, such as high deflection during bending tests, as well as control of crack propagation, has been achieved. The authors [8] noted that the developed segmented ceramic parts are excellent candidates for such critical applications, such as gas turbine combustion chambers, which have high requirements for heat resistance, good sound absorption, and high vibration resistance.

The quasi-deformation behavior of the segmented porous alumina ceramics was studied in our previous work [16]. Using the digital image correlation method, deformation distribution maps were obtained, which made it possible to observe deformation localization zones. At the first stage of compression tests, the deformation was localized on inhomogeneities, which led to the growth of the primary cracks. The next stage was characterized by emerging microcracks and fragmentation processes at the segment boundaries, accompanied with filling and compaction of the hollow spaces between the segments by the fragments followed by formation of compaction bands. These compaction bands significantly increased friction between the crack surfaces and delayed the primary crack propagation with simultaneous initiation of microcracking and microfragmentation between other segments until these compaction bands would occupy the whole volume of the sample. Such an inelastic behavior due to the formation of microcracks, fragmentation processes, and emerging bands of densified material ensured effective stress relaxation in porous, segmented alumina, as well as increasing its resistance to damage. Despite the results that have been obtained, some questions are still unanswered, in particular, how

the strength of the bond between the segments will affect the behavior during deformation and failure.

In this work, samples of segmented porous aluminum oxide ceramics with the same level of total porosity and the same segment sizes, but differing in the strength of the bond between the segments, were obtained, and their behavior during compression was studied by the DIC method.

## 2. Materials and Methods

### 2.1. Sample Preparation and Examination

A commercial alumina slip VK95-1 ("Contour" Cheboksary, Russia), consisting of 15 wt.% paraffin and 85 wt.% $\alpha$–$Al_2O_3$ with an average particle size of 4 µm, was heated above the melting point of paraffin (~ 80 °C) and mechanically mixed with 15 vol.% of spherical particles UHMWPE, with a size of 110 µm, until a homogeneous mixture was obtained, which was then used for injection molding and for obtaining cylindrical samples with a diameter of 10 mm and a height of 7 mm.

Paraffin evaporation was carried out by heating to 345 °C at heating rates of 0.35 °C/min (Sample #1), 0.48 °C/min (Sample #2), 0.56 °C/min (Sample #3), and 0.7 °C/min (Sample #4). UHMWPE was melted and then evaporated at the next stage by heating the samples to 600 °C at rates of 0.5 °C/min (Sample #1), 0.65 °C/min (Sample #4), 0.78 °C/min (Sample #3), 0.9 °C/min, and 1.0 °C/min (Sample #4). The oxygen contained in the alumina powder filler oxidized the evaporated paraffin and polymer to carbon dioxide, which then served to compact the filler. The isothermal stage involved heating all samples to 1000 °C at 1°C/min rate, followed by holding at this temperature for 30 min. After this heat treatment, samples contained no traces of organic compounds. Finally, sintering was carried out by heating the samples to 1300 °C for 3 h and holding at this temperature for another 1 h.

Then, samples were subjected to mechanical compression tests using a Devotrans GP 30 KN-DLC + CKS (Turkey) universal testing machine at a loading speed of $2 \times 10^{-4}$ s$^{-1}$. No lubrication was applied, so as to reduce the friction between the specimen ends and the machine's platens, except for aluminum foils, which served to protect the specimen's ends against fracture. Additionally, specimen height/diameter ratio was about 1:1, and, therefore, triaxial stress–strain state was established throughout the specimen volume and determined stress and strain localization. Three to five sample test samples were used to obtain reliable data on the compressive mechanical properties with respect to the selected modes of obtaining segmented ceramics. In other words, three to five test samples were used to mechanically validate each of the listed heating modes (Sample #1-#4).

The microstructure of the samples after sintering and compression tests was studied using a scanning electron microscopy SEM TESCAN VEGA 3 SBU (Tescan, Czech Republic).

### 2.2. DIC Procedure

A graphite powder was rubbed into the surface of the sintered porous samples to provide better visualization of the deformation pattern, to reduce the DIC error, and to control the ceramic microstructure during compression.

To carry out the DIC, a cylindrical surface was ground flat and polished to obtain a rectangular $7.2 \times 5.8$ mm flat field of interest (FOI). To detect possible micro- and macro-damages at different stages of deformation, samples with ground and then polished end faces were prepared. During the compression deformation, optical macro-images of the plane were taken every three seconds using a Nikon D90 camera (Nikon Corp., Tokyo, Japan) equipped with a macro lens and then saved on a hard drive.

The physical size of FOI ($7.2 \times 5.8$ mm) corresponded to the image size of $1890 \times 1535$ pixels. Photographing was carried out every 4.5 s, while the entire compression time was about 400 s at the compression speed of 0.1 mm/min. When calculating the displacement fields, the interval between the frames was changed from two to 10 frames. The parameters for calculating the displacement fields were set to $R = 131$–$185$ and $m = 115$–$125$, where $R$ is the size of the search area and $m$ is the pixel size of the reference (template) one. Here, the

parameter $m$ depends on the quality of the optical texture of the images themselves, and $R$ is determined by the range of the vector lengths at the edges of the vector field.

The measurement procedure was as follows. Two images were received from corresponding FOIs using the digital camera and then recorded on the computer hard disk. This process involved (1) the reference image (before mechanical loading) and (2) the current image (after loading).

The block algorithm was used to calculate the field of displacement vectors, as follows:

- The first image was divided into a number of template areas of $m \times m$ size, and the second image was divided into search areas of the $R \times R$ ($R > m$) size (Figure 1a).
- The reference area was then been scanned within the corresponding search area, and the value of the difference functional was calculated for each current position. In this case, scanning was performed line-by-line at a single pixel step.
- The desired vector was found from the coordinates of the global extremum of the functional shown in Figure 1b. A sub-pixel accuracy of the DIC results was achieved using a bicubic interpolation scheme for this functional.

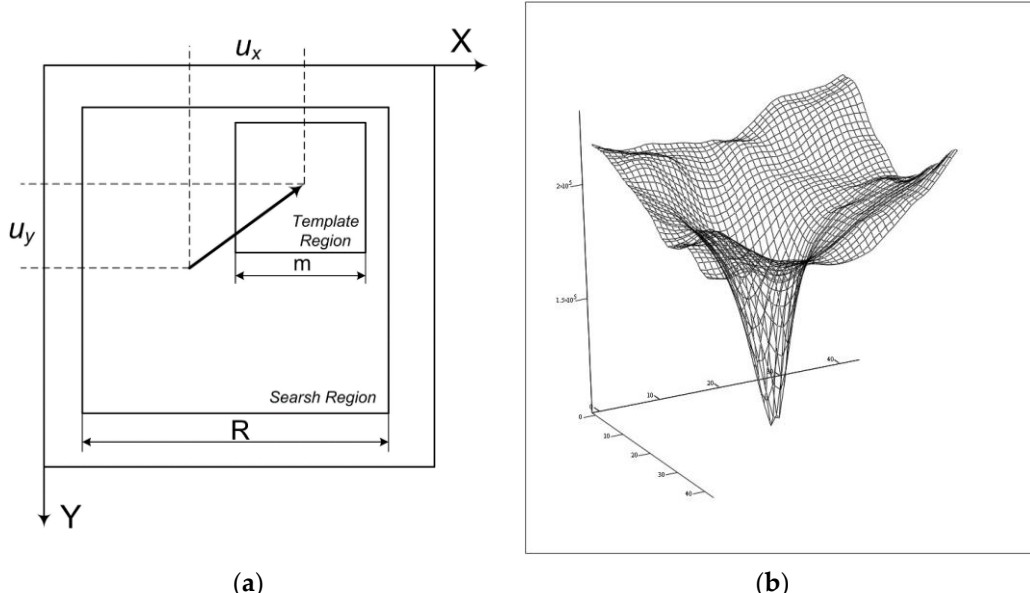

| (**a**) | (**b**) |

**Figure 1.** Scheme explaining the operation of the block identification algorithm (**a**) and a typical form of the difference functional (**b**).

The output data of the block algorithm were two arrays of displacement vector projections on the abscissa ($u_x(x, y)$) and ordinate axes ($u_y(x, y)$).

Let us assume that the displacement vector field computed from the images obtained in moments of time $t_1$ and $t_2$ corresponds to the current time $t = t_2$, while the material deformation act occurred during $\Delta t = t_2 - t_1$. The time $t$ corresponds to strain $\varepsilon$, so strain increment $\Delta\varepsilon$ is proportional to $\Delta t$.

The vector field is described by the relationship $\vec{u}(x, y) = u_x(x, y) \cdot \vec{e}_x + u_y(x, y) \cdot \vec{e}_y$, where $u_x(x, y)$—transverse, $u_y(x, y)$—longitudinal parts, and $\vec{e}_x$, $\vec{e}_y$—orts. The OY axis is directed along the external compression force. The type of deformation may be described as the cardinal plastic shear:

$$\gamma(x, y) = \sqrt{(\varepsilon_{xx} - \varepsilon_{yy})^2 + 4\varepsilon_{xy}^2} \tag{1}$$

where $\varepsilon_{xx}$, $\varepsilon_{yy}$, $\varepsilon_{xy}$—deformation tensor components.

This distribution can be mapped onto a pseudo-image in the following way. Let the deformation change in a certain interval, and let the brightness of the eight-bit image vary

in the range $0 \leq I(x, y) \leq 255$. By dividing the deformation interval into 255 parts and matching each of them to the corresponding brightness value, one obtains an eight-bit grayscale pseudo-image representing the deformation localization map. For an inverse pseudo-image, the darker the area of the image, the higher the deformation.

*2.3. Cardinal Plastic Shear*

Each vector field can be quantitatively related to the deformation averaged over the FOI area. Under conditions of active compression of the material at a constant speed, the mean deformation is generally homogeneous, unless some strain concentrator object, such as a crack, a local rotation mode, or a localized shear, does not interfere with such a scenario. If the deformation process proceeds uniformly, then the strain at each point of the sample should be the same value. This requirement is satisfied under the conditions that the dependences of the longitudinal and transverse displacements are linear functions of the coordinates.

$$u_x(x, y) = a_x \cdot x + b_x \cdot y + c_x$$
$$u_y(x, y) = a_y \cdot x + b_y \cdot y + c_y$$

$$(2)$$

$$\gamma = \sqrt{(\varepsilon_{xx} - \varepsilon_{yy})^2 + 4\varepsilon_{xy}^2} = \sqrt{(a_x - b_y)^2 + (b_x + a_y)^2}$$
$$\gamma_n = \gamma / (t_2 - t_1)$$

$$(3)$$

Here $a_x$, $b_x$, $c_x$, $a_y$, $b_y$, $c_y$ are the constants determined from experimental data, and $\gamma_n$ is the average specific deformation (cardinal plastic shear). Approximation was performed using the least squares method.

The degree of deviation of longitudinal or transverse displacements from the corresponding approximation plane is the standard deviation (SD), $\sigma_u$, normalized to the number of vectors of the given displacement field:

$$\sigma_u = \sqrt{D_x + D_y} / M \cdot N$$
$$\sigma_n = \sigma / \Delta t$$

where

$$\sigma = \sqrt{D_x + D_y}$$
$$D_x = \sum_{i=1}^{M} \sum_{j=1}^{N} ((u_x^a)_{i,j} - (u_x)_{i,j})^2$$
$$D_y = \sum_{i=1}^{M} \sum_{j=1}^{N} ((u_y^a)_{i,j} - (u_y)_{i,j})^2$$

$$(4)$$

$M(N)$ − the number of rows (columns) of arrays $u_x(x, y)$, $u_y(x, y)$, presented in discrete form. Here, the displacement standard deviation (SD), $\sigma_n$, is normalized to the time interval between two frames (or to the strain range). It is an important characteristic that makes it possible to single out the main stages of deformation and fracture and clarify their boundaries.

When calculating the vector fields, the time between frames was fixed (dn = 4, dn = 10). It can be seen that the dependence of the local deformation averaged over the area on the total deformation (strain) is non-linear. This, however, does not fundamentally change the variation of the time interval between frames. The SD is high when the mean strain itself is high or when we are approximating by a plane and the approximation error is significant.

When considering the behavior of the samples under consideration, the time interval was in the range of 2–10 frames. This scatter is due to the fact that, near the fracture, the flow velocity is very high, and, at a high time interval value, the error can become unacceptably high.

## 3. Results

*3.1. Structure of Sintered Samples*

Thermo-oxidization decomposition of pore-forming components made it possible to obtain a volume of gaseous products sufficient for the forming spherical pores, as

well as opening crack-like channels required for outgassing the samples. The rate of evaporation and outgassing of the blowing agent was controlled by the heating rates at both stages of outgassing, so that as the outgassing in green samples was slower, the heating rate to remove organic additives was lower. Final sintering at 1300 °C of the degassed samples resulted in partial sintering between the segments so that those pore channels became partially bridged (Figure 2a). A schematic of the segmented structure with the structural elements is shown in Figure 2b. On full sintering, the segments in all samples consisted of recrystallized 6–10 μm size grains with small 3 μm intergranular spherical pores (Figure 2b, 2) and large 110 μm size spherical pores (Figure 2b, 1). All samples had the same ~50% porosity. Pore channels (Figure 2b, 3) formed, in all four samples, a three-dimensional network that segmented sample volume (Figure 2b, 4) into ~230 μm.

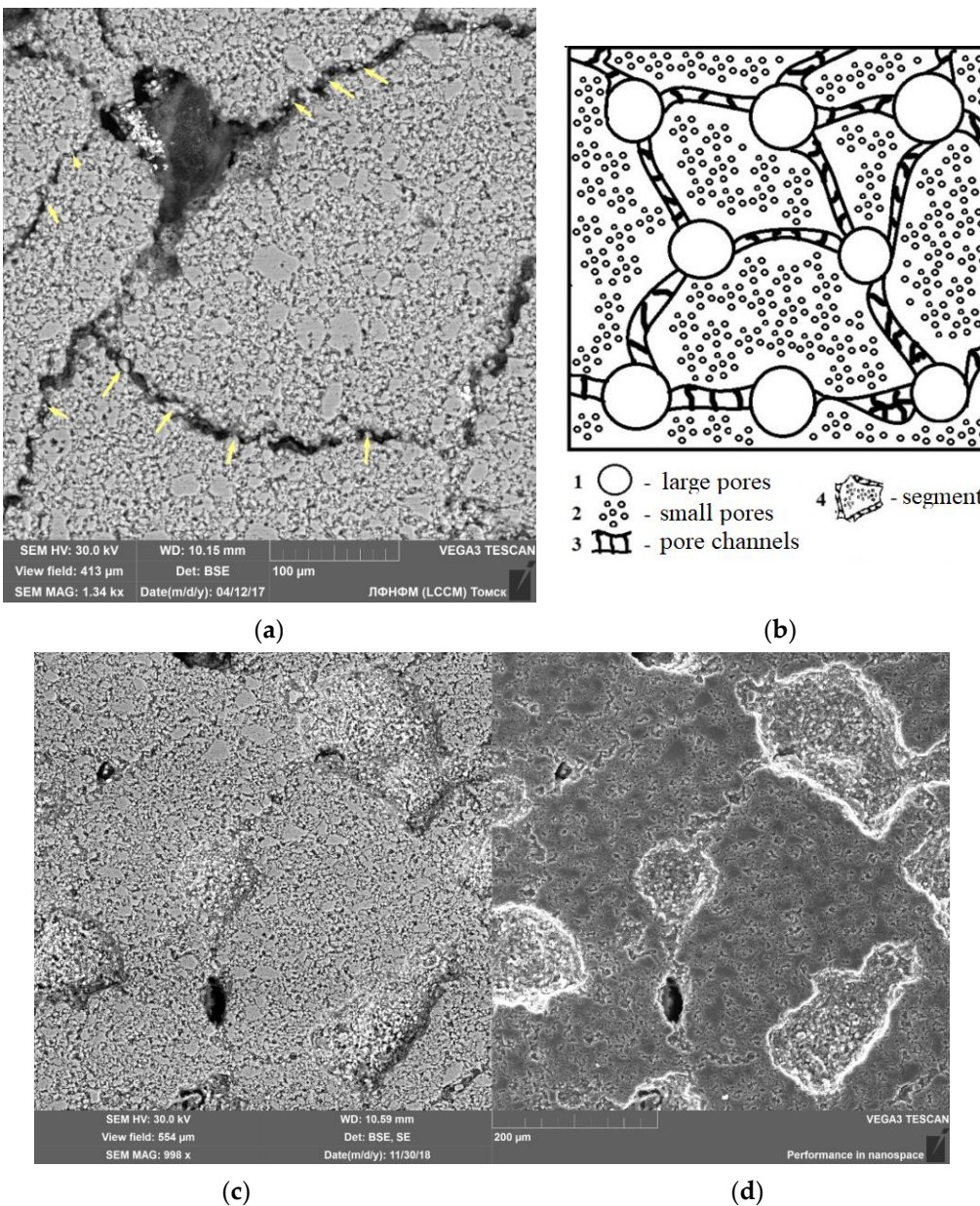

**Figure 2.** *Cont.*

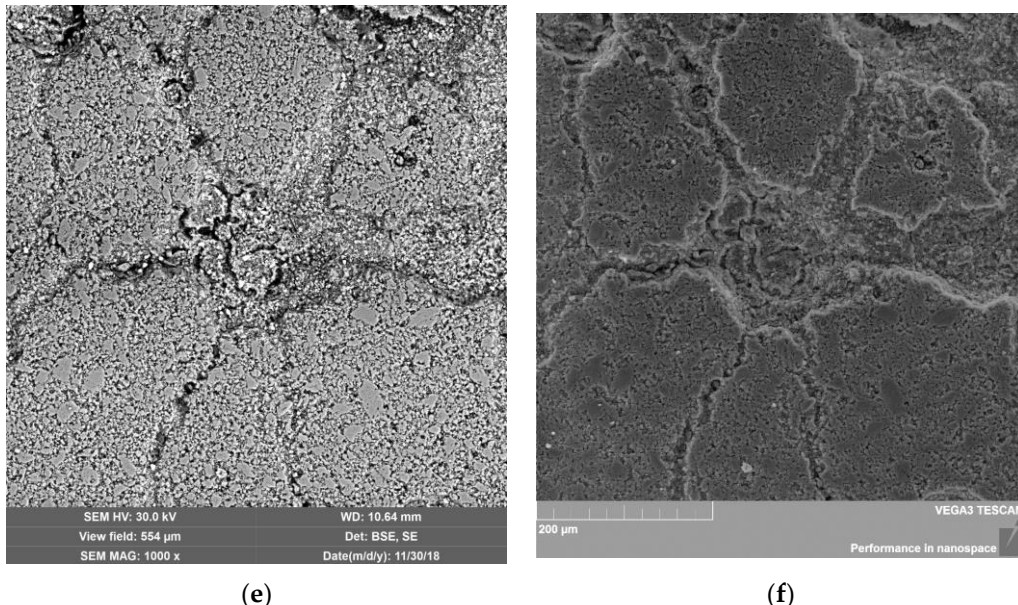

(**e**)                                                                    (**f**)

**Figure 2.** SEM BSE (**a**,**c**,**e**) and SE (**d**,**f**) images of as-sintered porous segmented ceramics. Scheme of the segmented structure with the indication of structural elements (**b**). Data for Sample #2 (**a**), Sample #1 (**c**,**d**), and Sample #4 (**e**,**f**) are shown.

The highest heating rate (Sample #4) resulted in forming ~20 μm-wide pore channels in sintered samples (Figure 2e,f), while the slowest heating (Sample #1) resulted in forming, by a factor of 2, narrower channels (Figure 2c,d).

### 3.2. Compression Test Results

Stress–strain diagrams in Figure 3a allow one to observe that the ultimate compressive strength (UCS) of samples subjected to outgassing is almost linearly reduced (Figure 3b) when the heating rate is increased, with widening of the intersegment channels. In addition, Figure 3a shows that strain-to-deformation value is increased for samples Sample #1, Sample #2 and Sample #3, while that of the Sample #4 curve is the minimum. Therefore, samples with the widest channels demonstrates the minimum UTS and tolerance to fracture.

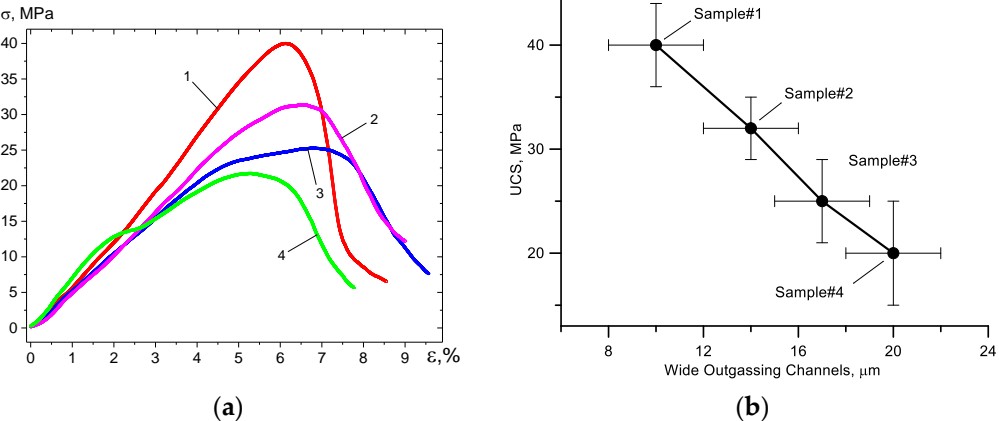

(**a**)                                                                    (**b**)

**Figure 3.** The monotonic compression "σ-ε" curves (**a**): 1—Sample #1; 2—Sample #2; 3—Sample #3; 4—Sample #4. UCS vs. wide pore channels curve (**b**).

The Sample #1 and Sample #4 compression behaviors will be discussed in detail below to gain further insight into deformation and fracture mechanisms inherent with such materials, as well as how to improve their design.

The difference in mechanical behavior under compression loading can be clearly observed from Figure 4 and Table 1, where sample #1's strength compares favorably with that of Sample #4. The presence of not-healed segment boundaries in Sample #4 has manifested itself in the non-linear first stage curve, as well as in the earlier onset of the post-fracture stage. The behavior of the strengthening factor $d\sigma/d\varepsilon$ allows even more clear delineation of the compression behavior stages (Figure 4b). In the first stage, $d\sigma/d\varepsilon$ values are positive and remain almost constant up to $\varepsilon = 0.06$ for Sample #1, while, for Sample #4, they are slightly reduced from the very beginning of the test up to the drop at $\varepsilon = 0.06$. In other words, the stress/strain curve allows observation of some inelastic behavior of the porous ceramics that is different from that of traditional brittle materials with catastrophic fracture (disintegration) after the nucleation of the very first crack.

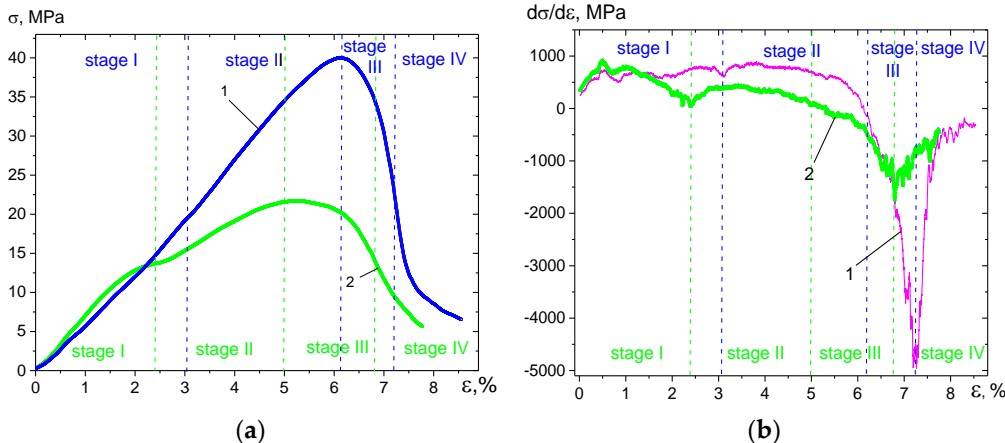

**Figure 4.** The monotonic compression "$\sigma$-$\varepsilon$" curve (**a**); the behavior of the "$d\sigma/d\varepsilon$-$\varepsilon$" curve (**b**). Sample #1 (1) and Sample #2 (2) data are shown.

**Table 1.** Mechanical properties of as-sintered porous segmented ceramics.

| Sample # | UCS, MPa | $\varepsilon_{max}$, % | $K = d\sigma/d\varepsilon$, MPa |
|---|---|---|---|
| 1 | 40 | 8.5 | $(-4800) < K < 860$ |
| 2 | 21 | 8.0 | $(-1600) < K < 890$ |

The post-fracture stages of Sample #1 and Sample #4 are characterized by "$d\sigma/d\varepsilon$-$\varepsilon$" curves, which sharply decrease down to their minimum negative values and again increase. In other words, the post-fracture behavior of segmented samples allows observation of strength degradation (pore collapse) [16] and strengthening (compaction band) stages [16].

Figure 4a and Table 1 show that it has ultimate compression strength (UCS) and maximum strain-to-fracture ($\varepsilon_{max}$) values higher than those of Sample #4.

Compressive UCS of Sample#1 was 40 MPa, and, in accordance with the results reported elsewhere [1], this value may be considered as satisfactory for 50% porous samples despite their segmented structure and network of partially healed segment boundaries.

More detailed DIC characterization of fracture and deformation in the segmented ceramics may be achieved by dividing both "$\sigma$-$\varepsilon$" and "$d\sigma/d\varepsilon$-$\varepsilon$" curves into four stages (Figure 4a,b).

Let Stage I exist in the ranges $(0.00 < \varepsilon \leq 0.03)$ and $(0.00 < \varepsilon \leq 0.025)$ for Sample #1 and Sample #4, respectively. This stage lasts from the onset of loading up to the first drop on the "$d\sigma/d\varepsilon$-$\varepsilon$" curve, which may be related to forming a primary crack. The next stage, Stage II, lasts from $(0.03 < \varepsilon \leq 0.062)$ to $(0.024 < \varepsilon \leq 0.050)$ for Sample #1 and #2, respectively, i.e., up to reaching the ultimate compression strength. Stage III corresponds to strain ranges $(0.072 < \varepsilon \leq 0.085)$ and $(0.68 < \varepsilon \leq 0.08)$, which reach the minimal $d\sigma/d\varepsilon$ values. Finally, Stage IV manifests full disintegration of the samples and densification of the remainder pile.

### 3.3. Deformation Maps

3.3.1. Stage I—Sample #1 ($0.00 < \varepsilon \leq 0.03$), Sample #4 ($0.00 < \varepsilon \leq 0.024$)

Despite the fact that optical micrographs of both samples in loading at Stage I look identical (Figures 5a,b and 6a,b), the corresponding pseudo-images allow observation of some deformation events on them (Figures 5c–f and 6b–f). Stage I may be characterized by emerging signs of strain localization that become observable at $\varepsilon \approx 0.018$ on sample 1 in the form of a vertical primary macroband (Figure 5c). At the same time, no changes can be observed on the corresponding optical micrographs (Figure 5a).

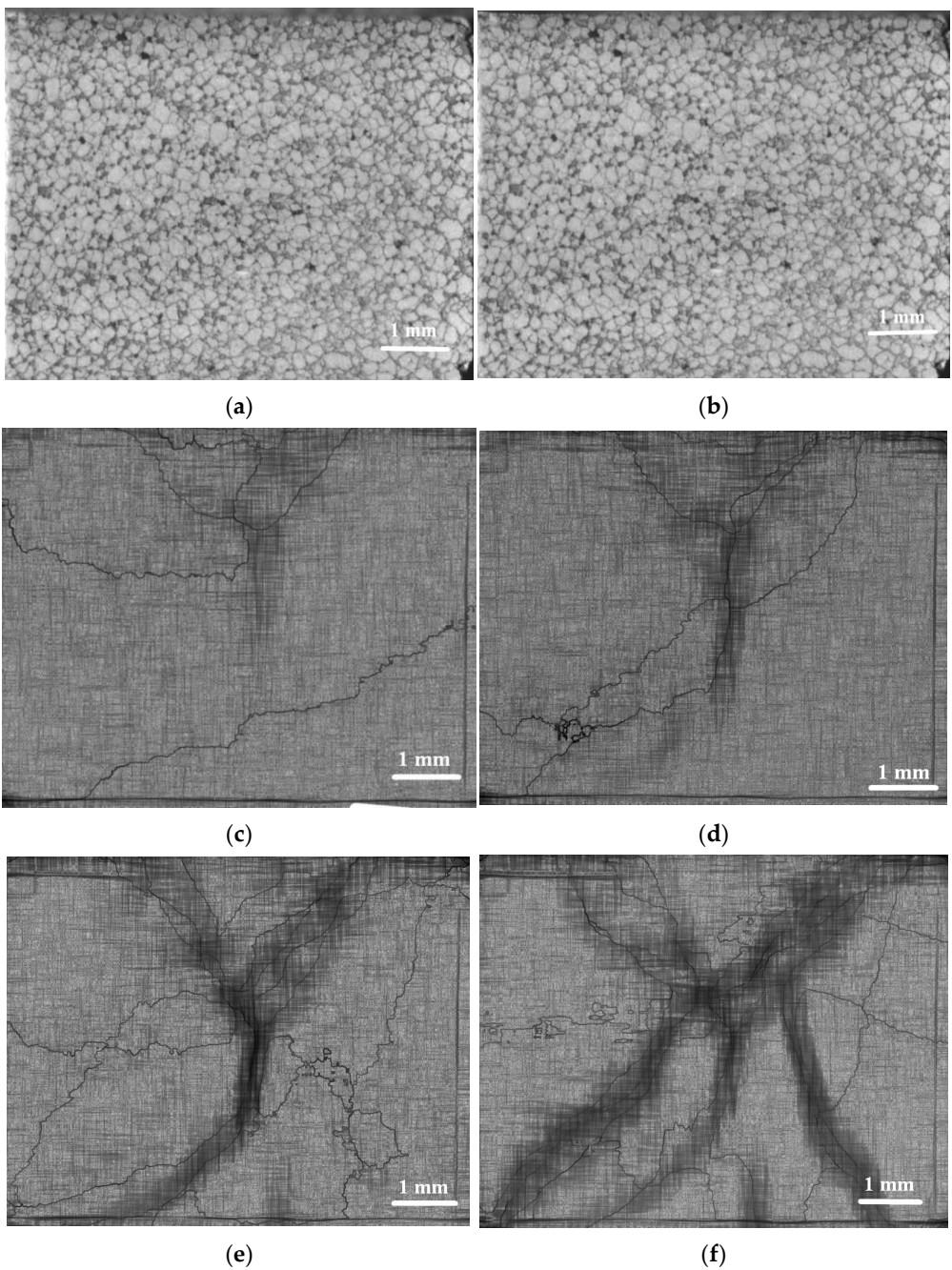

**Figure 5.** Optical macrographs (**a,b**) and deformation maps (**c–f**) of Sample #1 at the deformation Stage I at strain are as follows: $\varepsilon = 0.000$ (**a**), $\varepsilon = 0.030$ (**b**), $\varepsilon = 0.018$, $\Delta\varepsilon = 0.010$ (**c**), $\varepsilon = 0.021$, $\Delta\varepsilon = 0.010$ (**d**), $\varepsilon = 0.027$, $\Delta\varepsilon = 0.010$ (**e**), and $\varepsilon = 0.030$, $\Delta\varepsilon = 0.010$ (**f**).

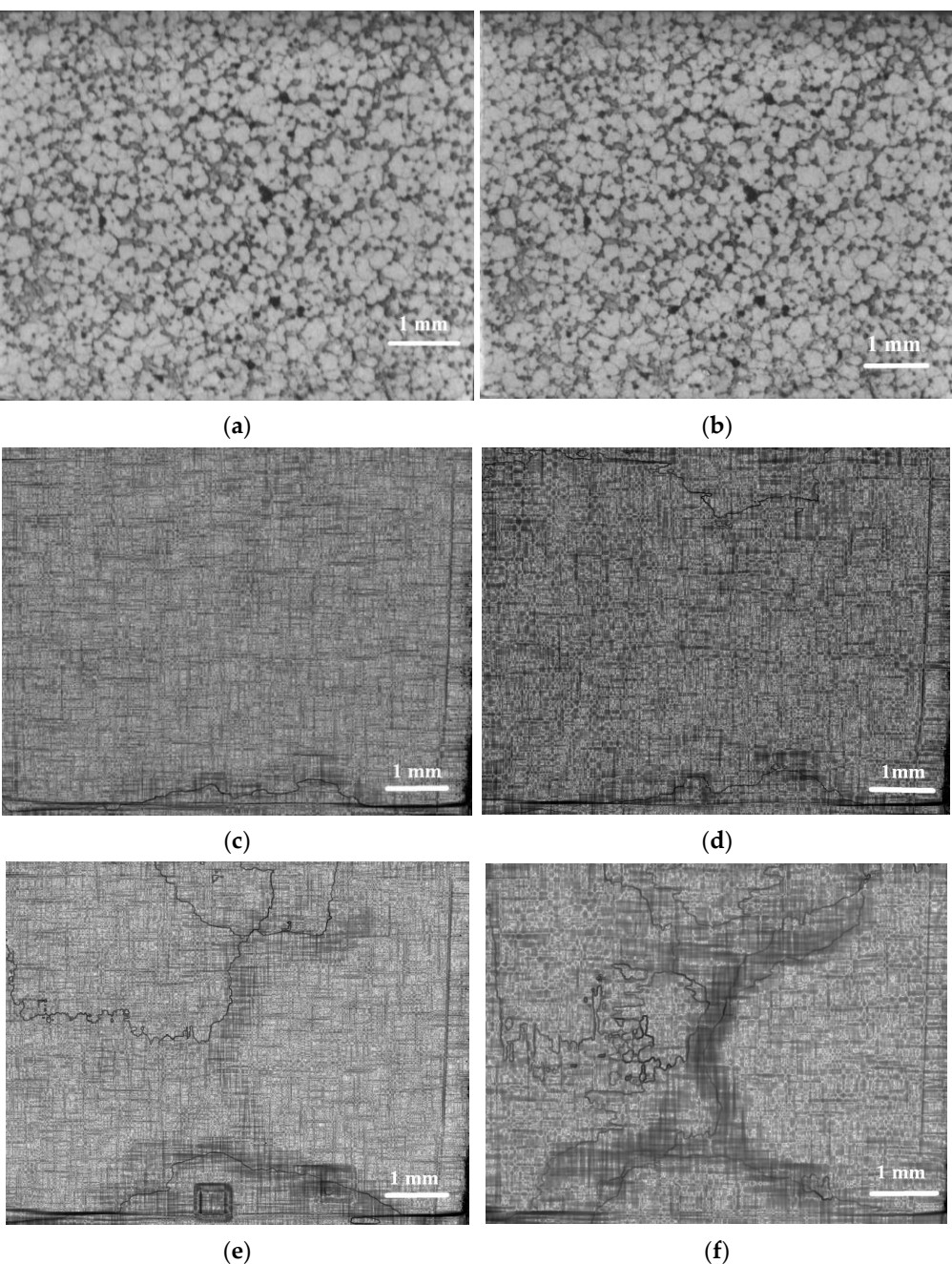

**Figure 6.** Optical macrographs (**a**,**b**) and deformation maps (**c**–**f**) of Sample #4 at the deformation Stage I at strain are as follows: $\varepsilon = 0.000$ (**a**), $\varepsilon = 0.024$ (**b**), $\varepsilon = 0.0076$, $\Delta\varepsilon = 0.0025$ (**c**), $\varepsilon = 0.011$, $\Delta\varepsilon = 0.0025$ (**d**), $\varepsilon = 0.015$, $\Delta\varepsilon = 0.0038$ (**e**), and $\varepsilon = 0.024$, $\Delta\varepsilon = 0.0063$ (**f**).

It should be noted that these primary cracks can be poorly seen due to existing networks of segment boundaries (Figures 5b and 6b). Further loading results in the generation of additional strain localization macrobands (SLMBs) deviating from the vertical line orientation (Figure 5d–f). At strain $\varepsilon \approx 0.03$, which corresponds to the end of Stage I, these SLMBs cover the total sample area.

Inelastic strain localization in Sample #4 starts at $\varepsilon \approx 0.0076$ when local stress concentration region is created in the center of the bottom part of the sample (Figure 6c). Then, a vertical macroband is formed in compression at $\varepsilon \approx 0.015$–0.025 (Figure 6e). Figure 7f shows that almost the entire cross section area of Sample #4 at $\varepsilon = 0.025$, $\Delta\varepsilon = 0.0063$ is shaded, implying the presence of numerous strain localization spots. In other words, under

these conditions, the deformation is almost inhomogeneous, possibly reflecting the process of multiple microcracking of the ceramics along the segment boundaries.

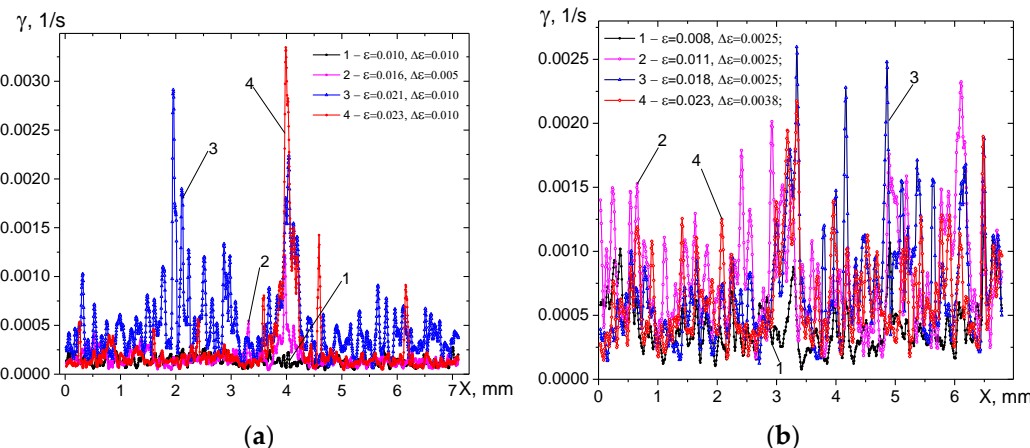

**Figure 7.** Spatial distribution of CPS rate $\gamma$ (x, y = $Y_m/2$), measured along the midline of Specimen #1 (**a**) and Specimen #4 (**b**) at the deformation Stage I.

Figure 7 shows the CPS distribution along the x axis. The multiple local maxima are formed throughout the sample section already at small strain (at $\varepsilon = 0.021$ in Sample #1, Figure 7a; at $\varepsilon = 0.011$ in Sample #4, Figure 7b). It can be seen that, in the case of Sample #1, the number of local CPS maxima is substantially lower than that of Sample #4, and their location corresponds to emerging SLMBs (Figure 5c–f).

### 3.3.2. Stage II—Sample #1 $(0.030 < \varepsilon \le 0.062)$, Sample #4 $(0.024 < \varepsilon \le 0.050)$

The primary cracks that formed in Sample #1 and Sample #4 can be observed on optical micrographs as bright areas (Figures 8a and 9a). Moreover, additional SLMBs are generated in Sample #1 and Sample #4 as strain increases (Figures 8b–d and Figures 9b–e).

Figure 10 shows that the intensity of the CPS local maxima of Sample #1 are about 2.5 times higher than that of Sample #4, i.e., the situation is similar to that occurring at Stage I. That is primary cracks continue growing in Sample #1, as demonstrated by high and narrow CPS peaks (Figure 10a, curves 2 and 3). However, these peaks are at least higher by a factor of two than those in Figure 7a, i.e., inelastic deformation by cracking became more intensive at this stage.

### 3.3.3. Stage III—Sample #1 $(0.062 < \varepsilon \le 0.072)$, Sample #4 $(0.050 < \varepsilon \le 0.068)$

Compression of Sample #1 at Stage III allows observation of almost the same crack structure (Figure 11a) as that shown previously in Figure 8d. Further loading did not result in changing this fracture pattern. At the same time, the width of the macrobands decreased with increase in the strain, from $\varepsilon = 0.064$ to $\varepsilon = 0.072$ (Figure 11b–d), so that they were turning into thin lines. Sample#4 demonstrated the same tendency (Figure 12b–d).

Figure 13 shows that the CPS maxima in Sample #1 became lower than those in Figure 10a, i.e., there was some mechanism that allowed retarding or even closing the primary cracks. It is important to note that CPS peaks in Sample #1 are approximately 10 times smaller than CPS peaks in Sample #4.

The CPS peaks in Sample #4 are still significantly lower than those of Sample #1, and their location corresponds to the emerging macrocracks that cross the sample from top to bottom and approximately coincide in direction with the compression axis (Figure 13). It is worthwhile to note that some CPS peaks became wider at this stage, as compared to those of the ones observed above at stages I and II. Such a fact could be implicit evidence of the inelastic strain developed by microcracking, fragmentation, and fragment displacement in the vicinity of segment boundaries, i.e., a mechanical energy dissipation mechanism served for crack arresting by high friction between the compacted fragments.

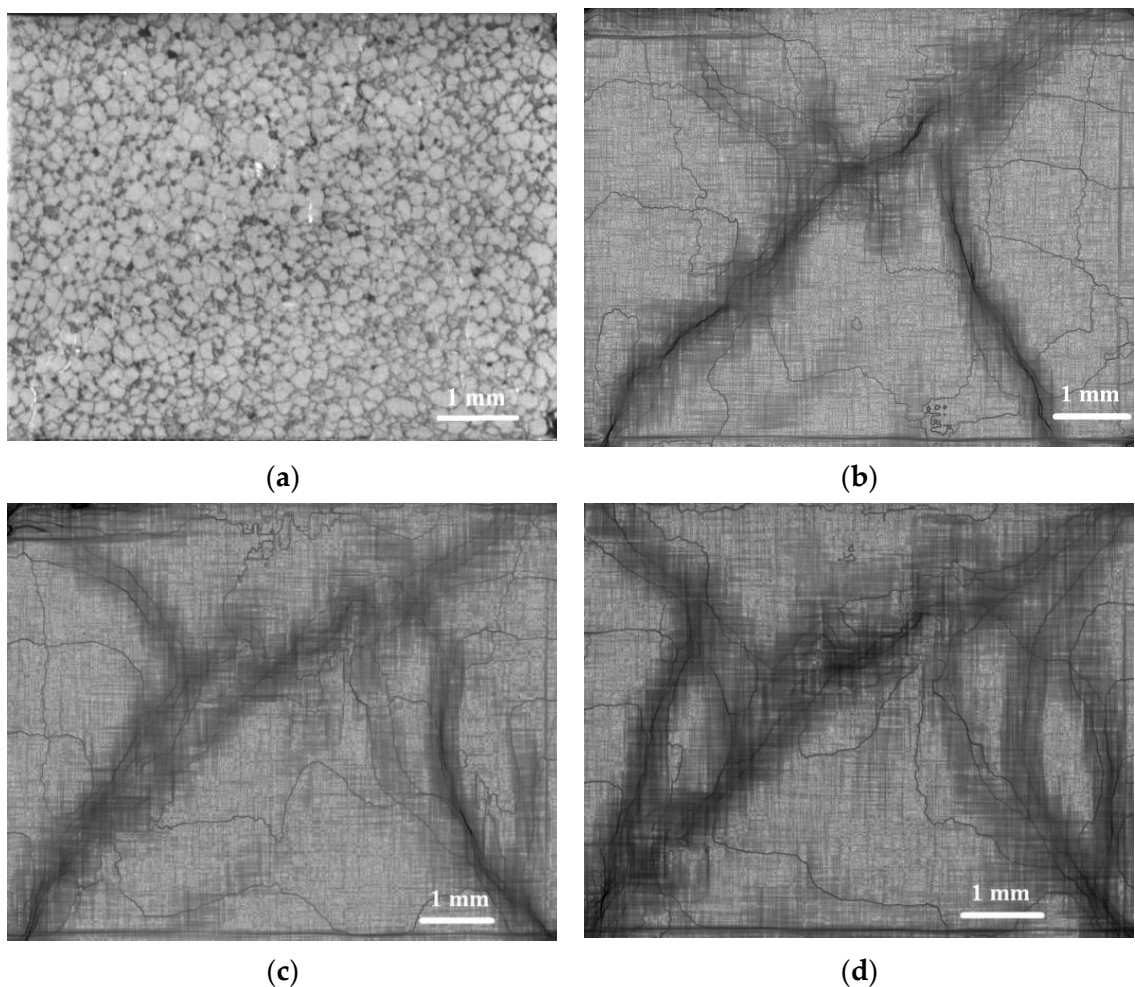

**Figure 8.** Optical macrograph (**a**) and deformation maps (**b**–**d**) of Sample #1 at the deformation Stage II at strain are as follows: $\varepsilon = 0.062$ (**a**), $\varepsilon = 0.037$, $\Delta\varepsilon = 0.010$ (**b**), $\varepsilon = 0.050$, $\Delta\varepsilon = 0.010$ (**c**), and $\varepsilon = 0.062$, $\Delta\varepsilon = 0.010$ (**d**).

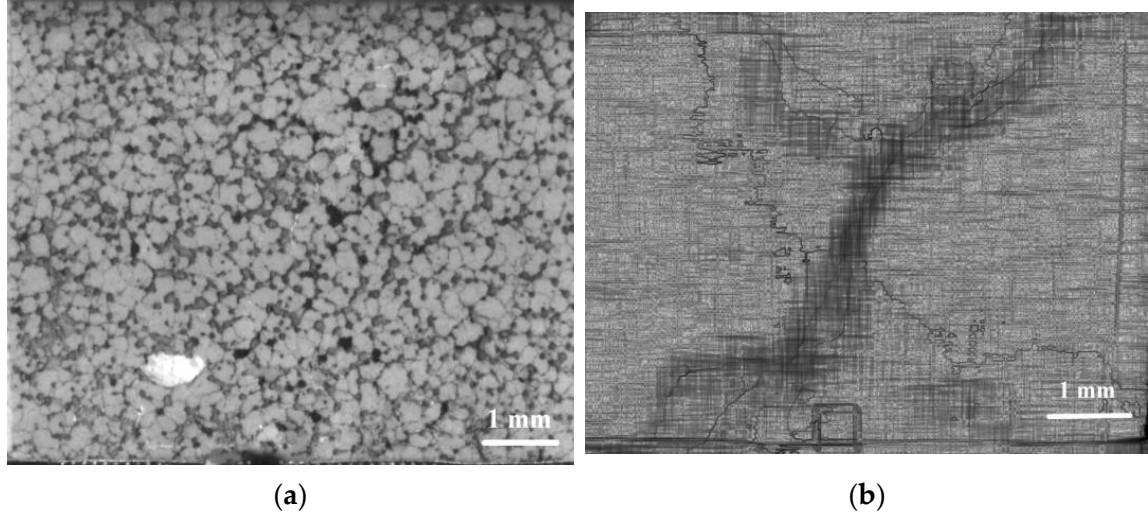

**Figure 9.** *Cont.*

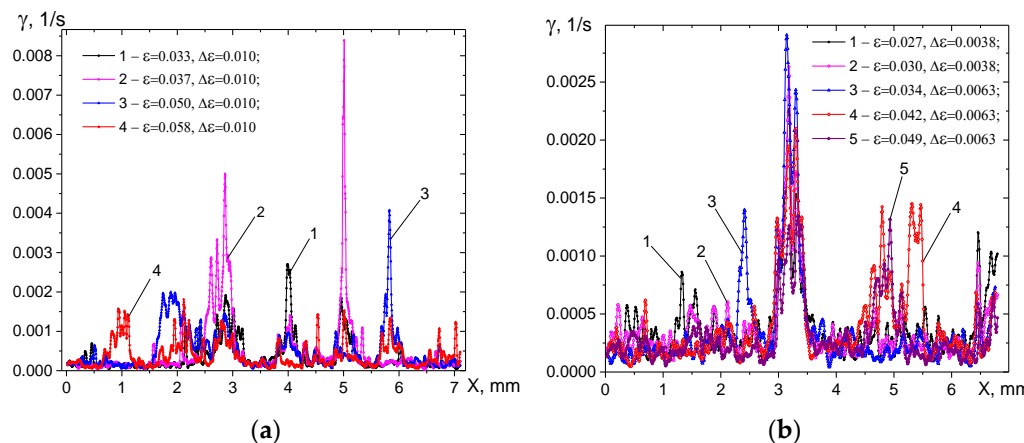

(**c**) (**d**)

(**e**)

**Figure 9.** Optical macrograph (**a**) and deformation maps (**b-e**) of Sample #4 at the deformation Stage II at strain are as follows: $\varepsilon = 0.050$ (**a**), $\varepsilon = 0.034$, $\Delta\varepsilon = 0.0063$ (**b**), $\varepsilon = 0.042$, $\Delta\varepsilon = 0.0063$ (**c**), $\varepsilon = 0.046$, $\Delta\varepsilon = 0.0063$ (**d**), and $\varepsilon = 0.050$, $\Delta\varepsilon = 0.0063$ (**e**).

(**a**) (**b**)

**Figure 10.** Spatial distribution of CPS rate $\gamma$ (x, y = $Y_m/2$), measured along the midline of Specimen #1 (**a**) and Specimen #4 (**b**) at the deformation Stage II.

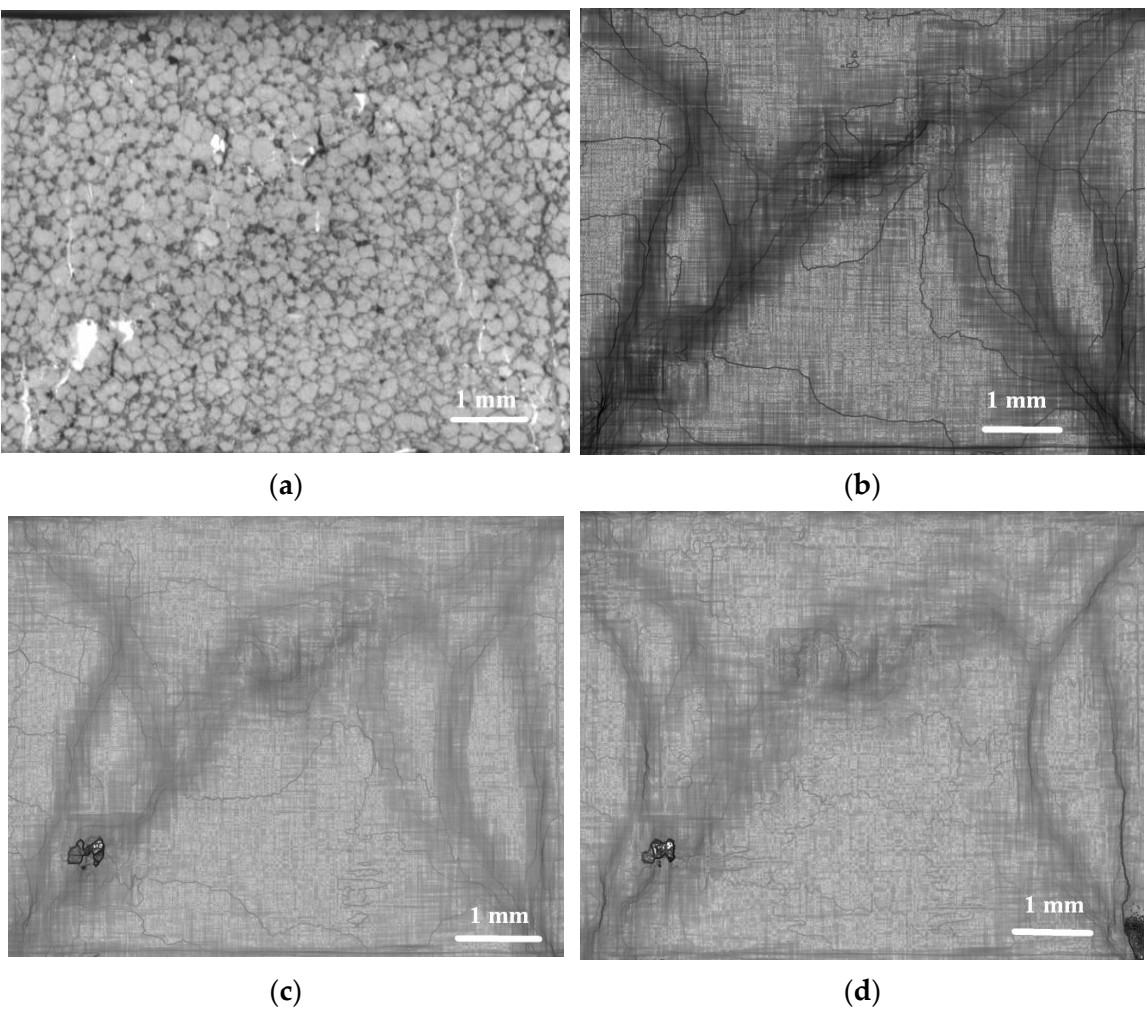

**Figure 11.** Optical macrograph (**a**) and deformation maps (**b–d**) of Sample #1 at the deformation Stage III at strain are as follows:$\varepsilon = 0.072$ (**a**), $\varepsilon = 0.064$, $\Delta\varepsilon = 0.010$ (**b**), $\varepsilon = 0.067$, $\Delta\varepsilon = 0.010$ (**c**), and $\varepsilon = 0.072$, $\Delta\varepsilon = 0.010$ (**d**).

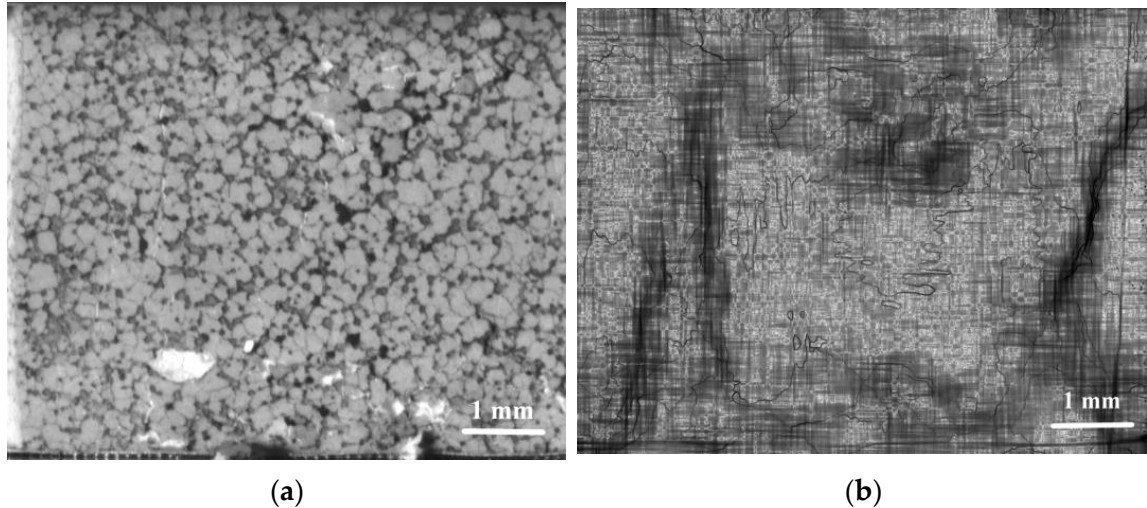

**Figure 12.** *Cont.*

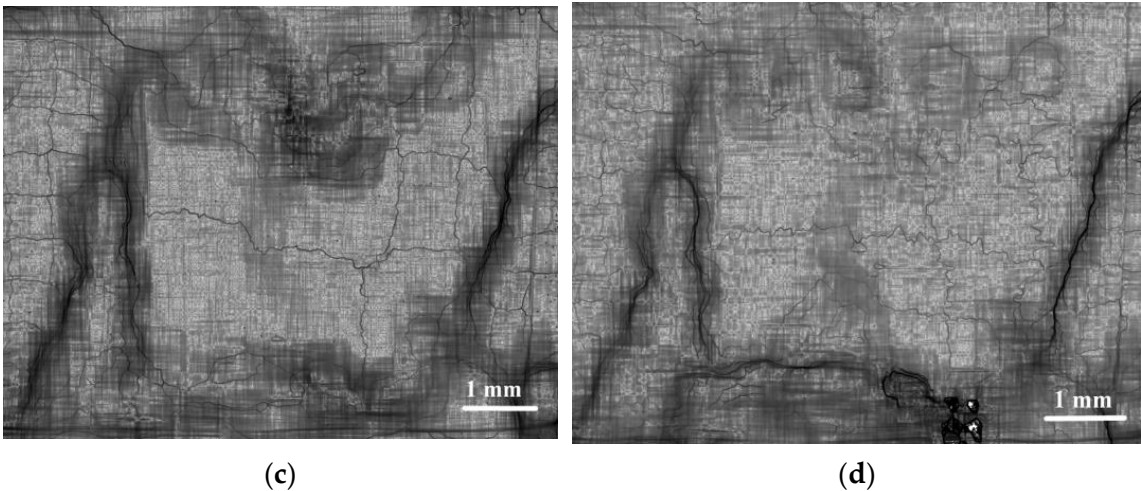

(**c**) (**d**)

**Figure 12.** Optical macrograph (**a**) and deformation maps (**b**–**d**) of Sample #4 at the deformation Stage III at strain are as follows: $\varepsilon = 0.068$ (**a**), $\varepsilon = 0.060$, $\Delta\varepsilon = 0.0063$ (**b**), $\varepsilon = 0.064$, $\Delta\varepsilon = 0.0063$ (**c**), and $\varepsilon = 0.071$, $\Delta\varepsilon = 0.0063$ (**d**).

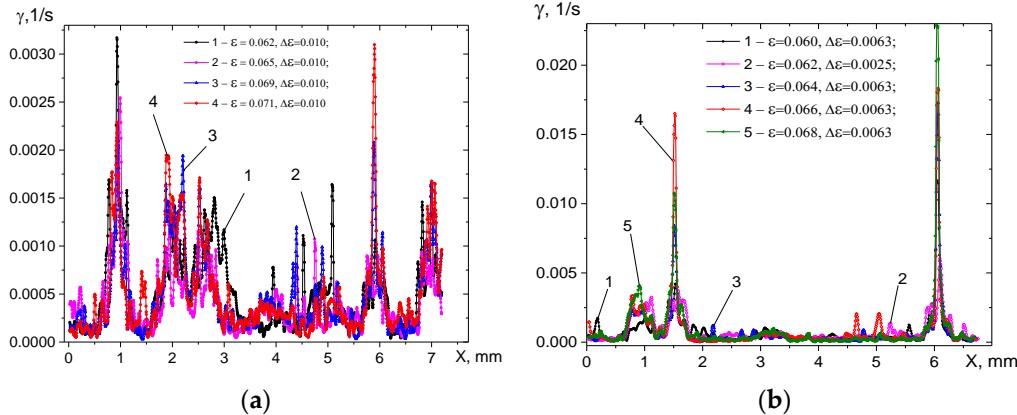

(**a**) (**b**)

**Figure 13.** Spatial distribution of CPS rate $\gamma$ (x, y = $Y_m/2$), measured along the midline of Specimen #1 (**a**) and Sample #4 (**b**) at the deformation Stage III.

3.3.4. Stage IV—Sample #1 $(0.072 < \varepsilon \leq 0.085)$, Sample #4 $(0.068 < \varepsilon \leq 0.080)$

At this stage, both Sample #1 and Sample #4 lose their integrity and break into individual fragments with macrocracks (Figures 14a and 15f). Corresponding pseudo-images show inelastic strain localization mainly in the form of thin lines (Figures 14b–d and 15b–d), whose location coincides with that of the macrocracks in Figures 14a and 15f.

At Stage IV, the CPS peaks in Sample #1 (Figure 16a) became approximately equal in value to the CPS peaks in Sample #4.

The observed fluctuation of CPS peaks on Sample #1 from stage to stage (Figures 7a, 10a, 13a and 16a) was in clear contrast to Sample #4, where the intensity of CPS peaks only increased with increasing strain (Figures 7b, 10b, 13b and 16b).

*3.4. Mean Cardinal Plastic Shear Evolution with Strain*

The CPS and standard deviation dependencies vs. strain are shown in Figure 17. The CPS vs. strain dependence for Sample #1 shows a clear bump covering Stage I—within the $\varepsilon = 2–5\%$ range—while Sample #4 demonstrates almost monotonous CPS growth with the strain (Figure 17a). When looking into how the CPS was accumulating in this strain range, one can see that the highest, except for those at Stage IV, CPS peaks were observed in this strain range (Figure 10a) that contributed to the mean CPS.

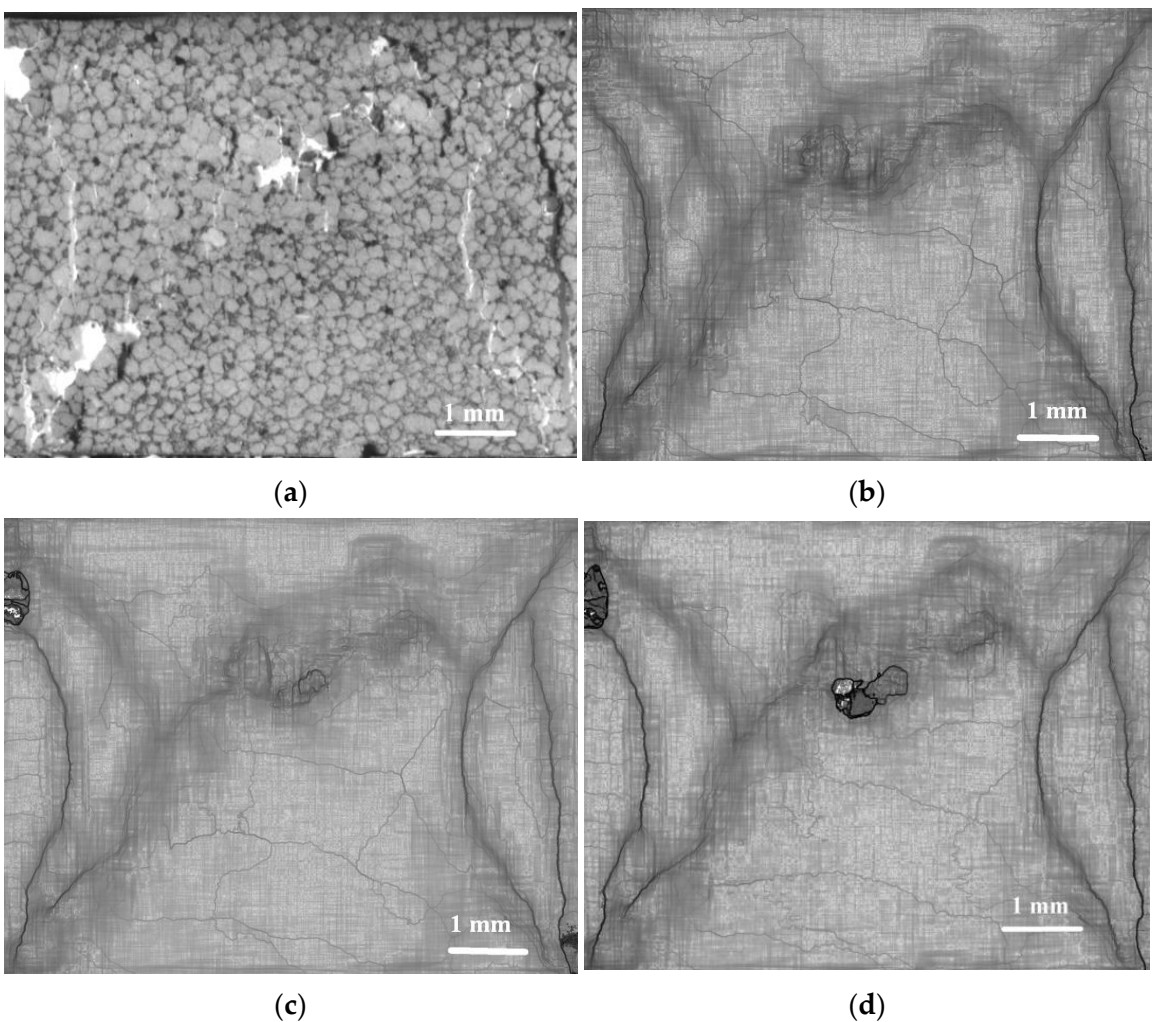

**Figure 14.** Optical macrograph (**a**) and deformation maps (**b**–**d**) of Sample #1 at the deformation Stage IV at strain are as follows: $\varepsilon = 0.085$ (**a**), $\varepsilon = 0.077$, $\Delta\varepsilon = 0.010$ (**b**), $\varepsilon = 0.079$, $\Delta\varepsilon = 0.010$ (**c**), and $\varepsilon = 0.081$, $\Delta\varepsilon = 0.010$ (**d**).

The similar tendency can be followed from the standard deviation vs. strain dependency (Figure 17b), which has even more sensitivity to the inelastic deformation events. In particular, the onset portion of the sample 1 and 2 curves (Stage I) allow the observation of small peaks. These peaks correspond to elevated mean CPS and standard deviation values, which, as proposed, may be related with forming an initial crack in both materials. Additionally, the Sample #4 curve is characterized by the presence of small peaks and troughs that may reflect on the cyclic character of inelastic deformation by cracking, followed by accumulation of stress and cracking again.

### 3.5. Estimation of Microfracture-Retardation Mechanism by Rate-of-Strain Tensor (ROST) Components

The evolution of the ROST components in both samples along the corresponding coordinate axes, as depended on the total strain, is represented in Figures 18 and 19. The Sample #1 ROST component $\dot{\varepsilon}_{xx}(X, Y = Y_m/2)$ demonstrates a nonlinear crack propagation when it starts opening at low strain up to its maximum at $\varepsilon = 4\%$ and then starts closing to its full disappearance at the end of the diagram (Figure 18a). The Sample #4 ROST component $\dot{\varepsilon}_{xx}(X, Y = Y_m/2)$ implies continuous cracking with the strain (Figure 19a). The $\dot{\varepsilon}_{yy}(Y, X = X_m/2)$ ROST components for Sample #1 and Sample #4 reveal similar behavior and may be related to displacement, microcracking, and fragmentation at the crack edges (Figures 18b and 19b).

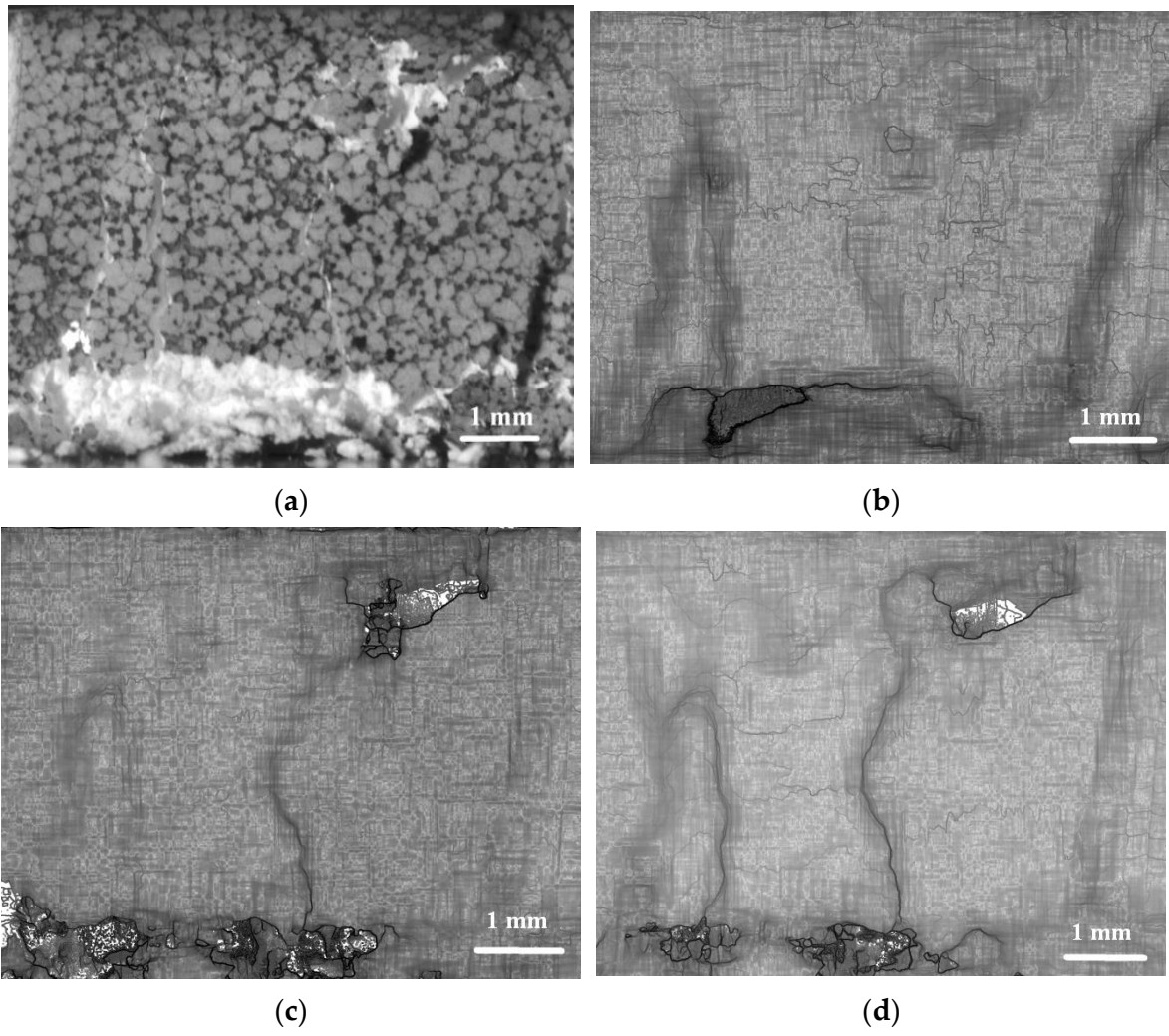

**Figure 15.** Optical macrograph (**a**) and deformation maps (**b**–**d**) of Sample #4 at the deformation Stage IV at strain are as follows: $\varepsilon = 0.080$ (**a**), $\varepsilon = 0.076$, $\Delta\varepsilon = 0.0025$ (**b**), $\varepsilon = 0.079$, $\Delta\varepsilon = 0.0038$ (**c**), and $\varepsilon = 0.080$, $\Delta\varepsilon = 0.0013$ (**d**).

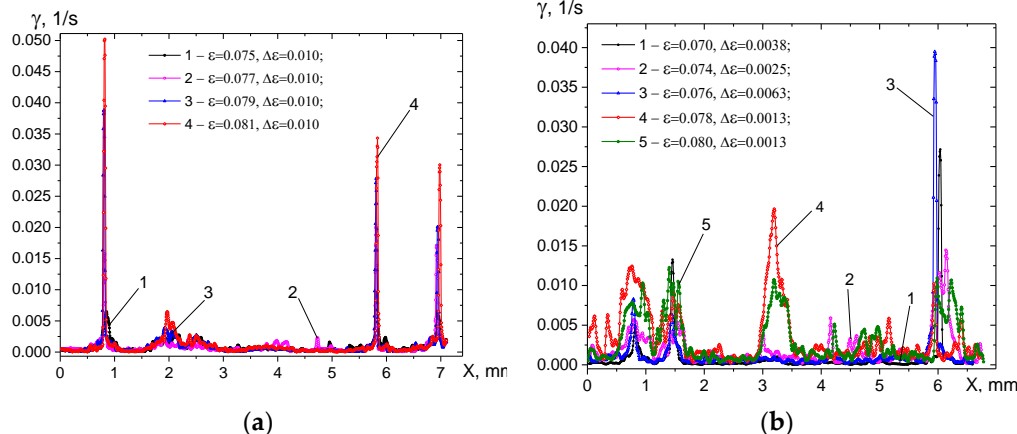

**Figure 16.** Spatial distribution of CPS rate $\gamma$ (x, y = $Y_m/2$), measured along the midline of Specimen #1 (**a**) and Specimen #4 (**b**) at the deformation Stage IV.

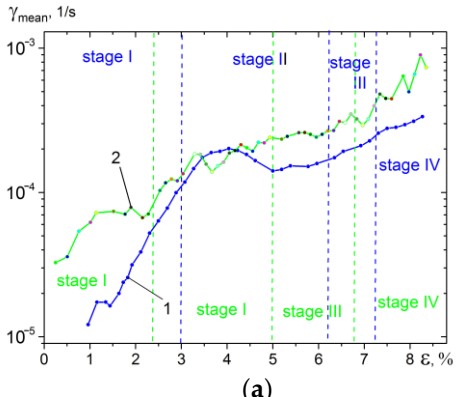
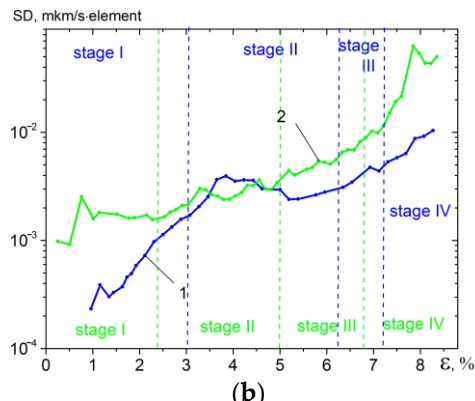

**Figure 17.** Dependencies of mean CPS vs. strain $\varepsilon$ (**a**), standard deviations vs. strain $\varepsilon$ (**b**). Sample #1 (1) and Sample #4 (2) data are shown.

Based on the DIC dates, the full-field displacement distribution can be obtained, and then the strain distributions and the average strains of the vertical direction (loading direction) and transverse direction (vertical to the loading direction) are calculated [17]. Additionally, the Poisson ratio (in-plane Poisson ratio) is the absolute value of the ratio of transverse average strain to vertical average strain [17].

The Poisson ratio was calculated according to a procedure the same as that used for calculating the cardinal plastic shear in Section 2.3 and using the vector fields obtained for T = 30, with the acquired frame period, the value was equal to that of 10 frames. The inelastic flow was assumed to be homogeneous and, therefore, both longitudinal and transversal displacements were approximated by planes. The value calculated according to such a procedure value was similar to the Poisson ratio by definition. The deformation components, however, were the non-recoverable strains averaged by the corresponding areas.

As can be seen in Figure 20, starting from small values of $\varepsilon$, the value of the Poisson ratio increased under compression, and since it is a characteristic of an elastic material, the increase in the ratio $\varepsilon_{xx}/\varepsilon_{yy}$ is higher than 0.29 (0.15–0.29 is the value of Poisson ratio $Al_2O_3$ with different levels of porosity and stiffness [18–20]), and, at $\varepsilon = 0.02$, this means that these test samples are deformed inelastically (demonstrated by the effect of pseudo-plasticity), with the maximum achieved for Sample #1 and #4 values of the ratio $\varepsilon_{xx}/\varepsilon_{yy}$ at Stage IV. It can be seen that the evolution of $n = \varepsilon_{xx}/\varepsilon_{yy}$ ratio with inelastic strain for Sample #1 and Sample #4 is similar to dependencies of mean CPS vs. strain (Figure 17a), with the maximum $\varepsilon_{xx}/\varepsilon_{yy}$ ratio achieved for Sample#1 at $\varepsilon = 0.04$ and an almost continuous increase in $\varepsilon_{xx}/\varepsilon_{yy}$ ratio with increasing strain occurs for Sample#4.

*3.6. SEM Post-Mortem Analysis*

Microcracking, fragmentation, and compacting at the segment boundaries may be illustrated using SEM BSE images in Figure 21, where wide compaction bands formed in them (Figure 21a,b) after filling the boundaries with 5–20 μm fragments (Figure 21a,c). These bands may be classified into long primary compaction bands, as are shown by the arrows in Figure 11a and accommodation compaction bands, which form around the segments (Figure 21b). It can be observed from Figure 21b,c that the large pores are partially or fully filled with the fragments.

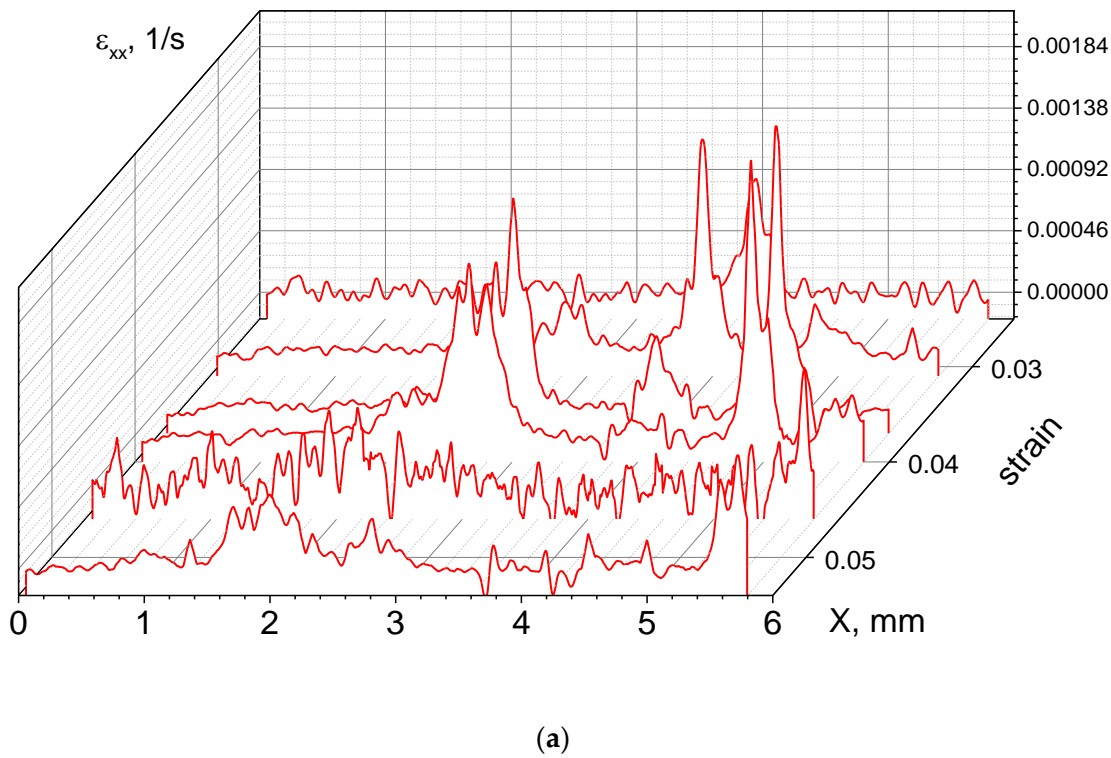

(**a**)

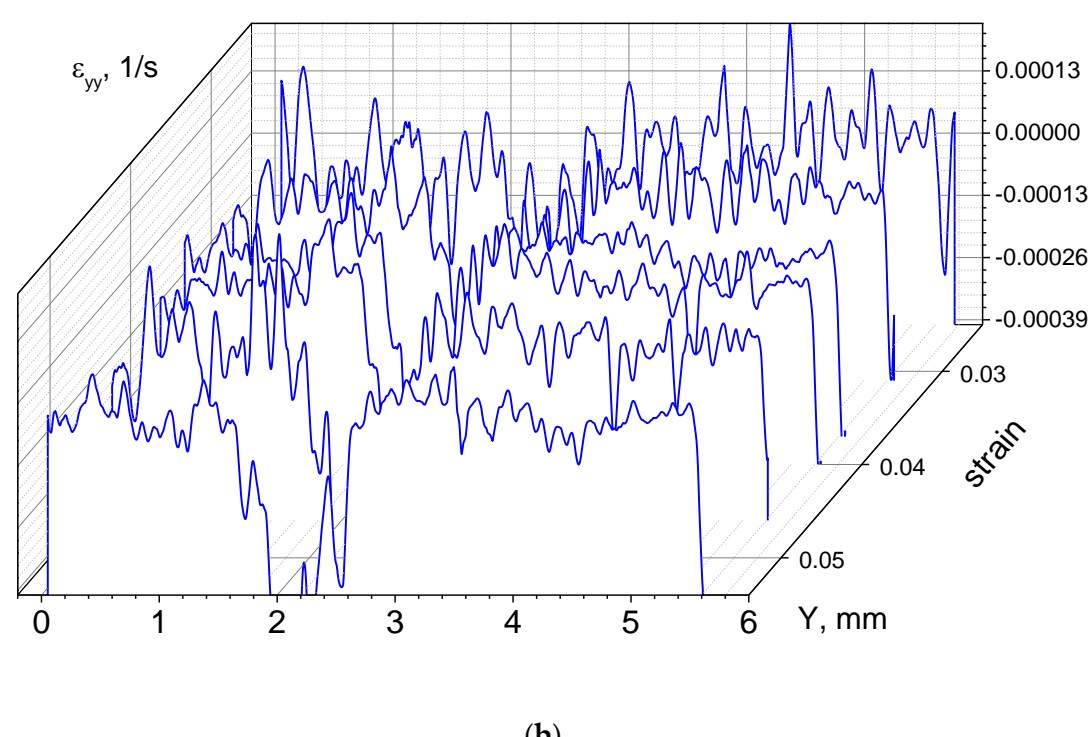

(**b**)

**Figure 18.** Evolution of ROST components $\dot{\varepsilon}_{xx}(X, Y = Y_m/2)$ (**a**) and $\dot{\varepsilon}_{yy}(Y, X = X_m/2)$ (**b**). Data from Sample #1 are shown.

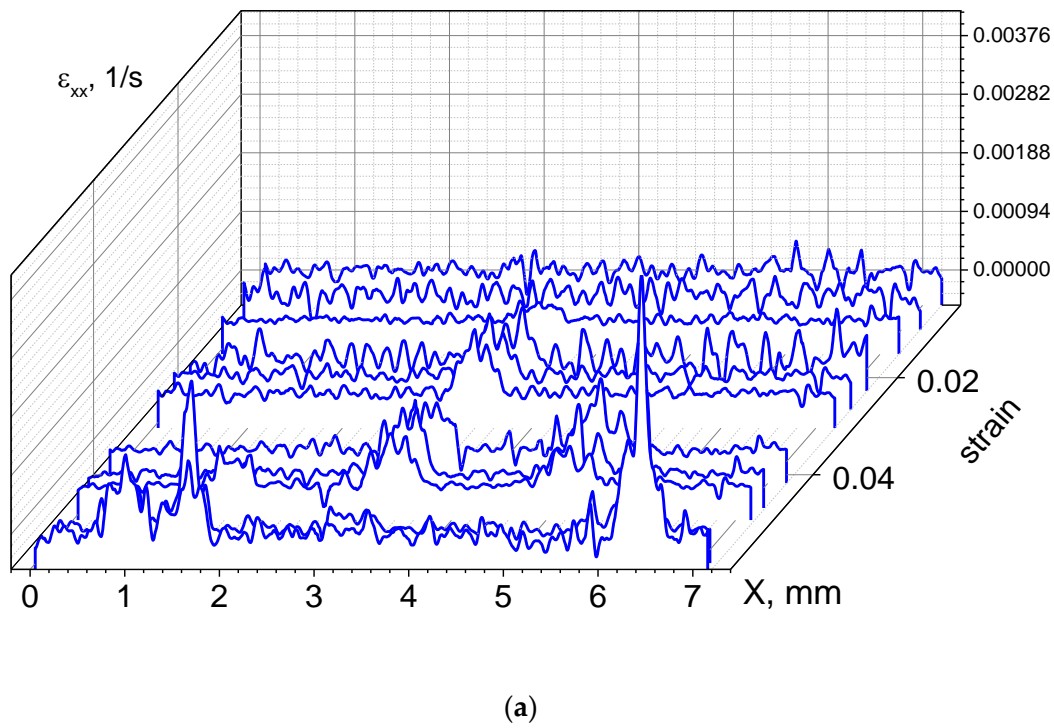

(**a**)

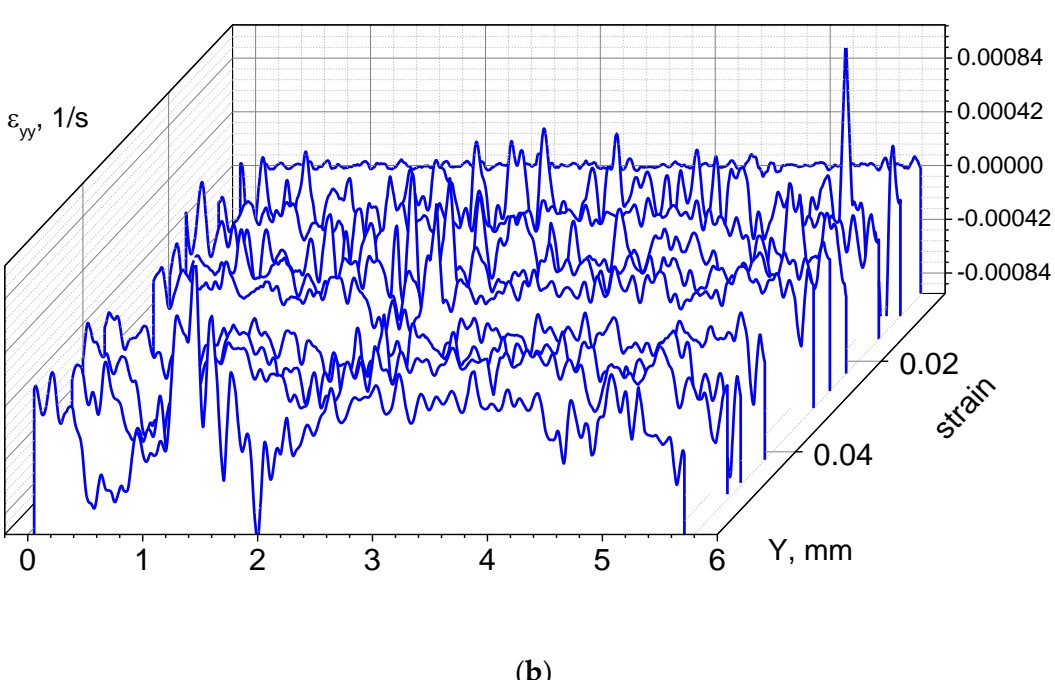

(**b**)

**Figure 19.** Evolution of ROST components $\dot{\varepsilon}_{xx}(X, Y = Y_m/2)$ (**a**) and $\dot{\varepsilon}_{yy}(Y, X = X_m/2)$ (**b**). Data from Sample #4 are shown.

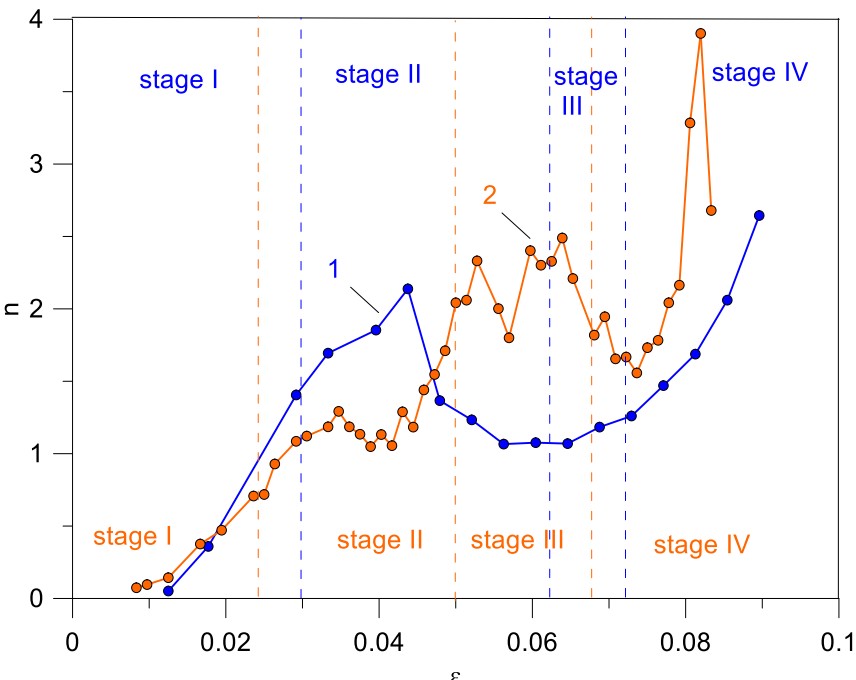

**Figure 20.** Evolution of $n = \varepsilon_{xx}/\varepsilon_{yy}$ ratio with inelastic strain. Sample #1 (1) and Sample #2 (2) data are shown.

The action of the crack retardation mechanism may be related to microcracking, fragmentation, and compaction of the intersegment spaces (discontinuities) and therefore can be illustrated using the SEM images where intersegment spaces are partially filled with fragments of material 10–20 μm in size (Figure 22). Figure 22a,b shows that the structure of Sample #1 is more compacted, with a smaller average size of segments/blocks formed as a result of compression, as well as narrow segment boundaries. The average size of the segments/blocks of Sample #1 was 100 μm (Figure 22a,b), while for Sample #4 it was about 190 μm (Figure 22c,d).

### 3.7. Circulation as a Characteristic of Vortex Flow

It was established above that inelastic deformation in both ceramic samples is performed both by fast fracturing along the stress concentrators, such as the segment boundaries, and by accumulation of displacements in the vicinity of these boundaries. The vector fields obtained from the porous, segmented alumina samples may show some sort of vortex patterns that we believe originate from these near boundary processes, including generation of the earlier described compaction bands (Figure 23). The reason behind such a correlation may be cooperation of numerous shear deformation events occurring on the segment boundaries at the fragmentation and compaction band generation stages, which result in mutual reorientation of the segments and pseudo-vortex rotation. Therefore, quantification of these deformation modes may serve for evaluating the intensity of correlated friction and energy absorbance processes on the segment boundaries, i.e., the high or low damage tolerance. The minimum rotation mode will mean low damage tolerance.

A convenient approach to describe the vortex motion may be using characteristics such as vorticity [21] and circulation ($\Gamma$):

$$\Gamma = \iint_S rot\vec{V}dS = \oint_L (\vec{V}\vec{dl}) \tag{5}$$

Here, $\vec{V}(x,y)$ is the flow velocity on the plane, and $S$ is the area of the region bounded by the contour of length, $L$. Equation (5) is based on the Stokes theorem. If the circulation is not equal to zero, then there is a vortex flow in the given area.

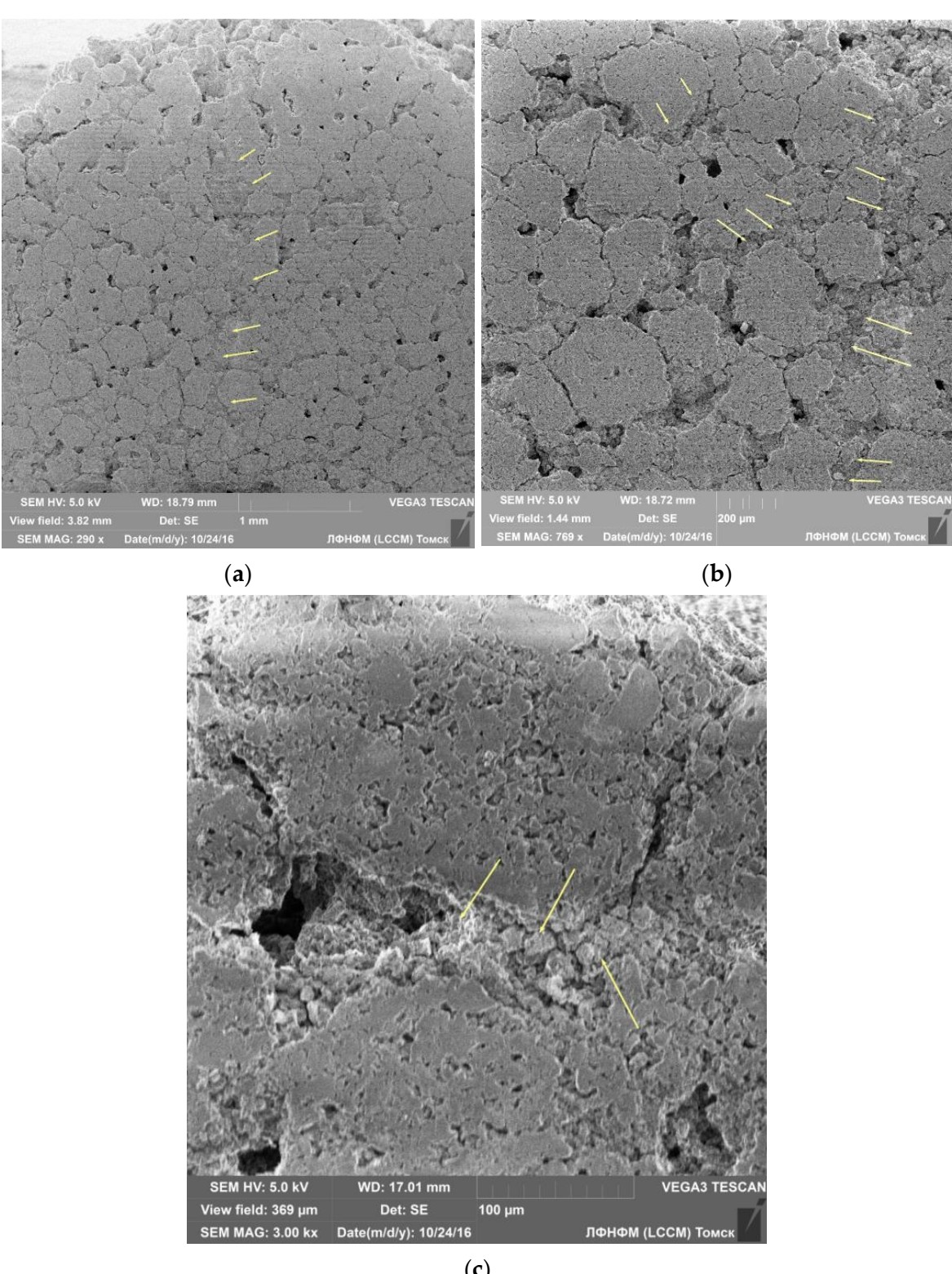

**Figure 21.** Ex situ SEM BSE images of compaction macrobands (**a**) and accommodative compaction bands (**b**,**c**). Sample #1 data are shown.

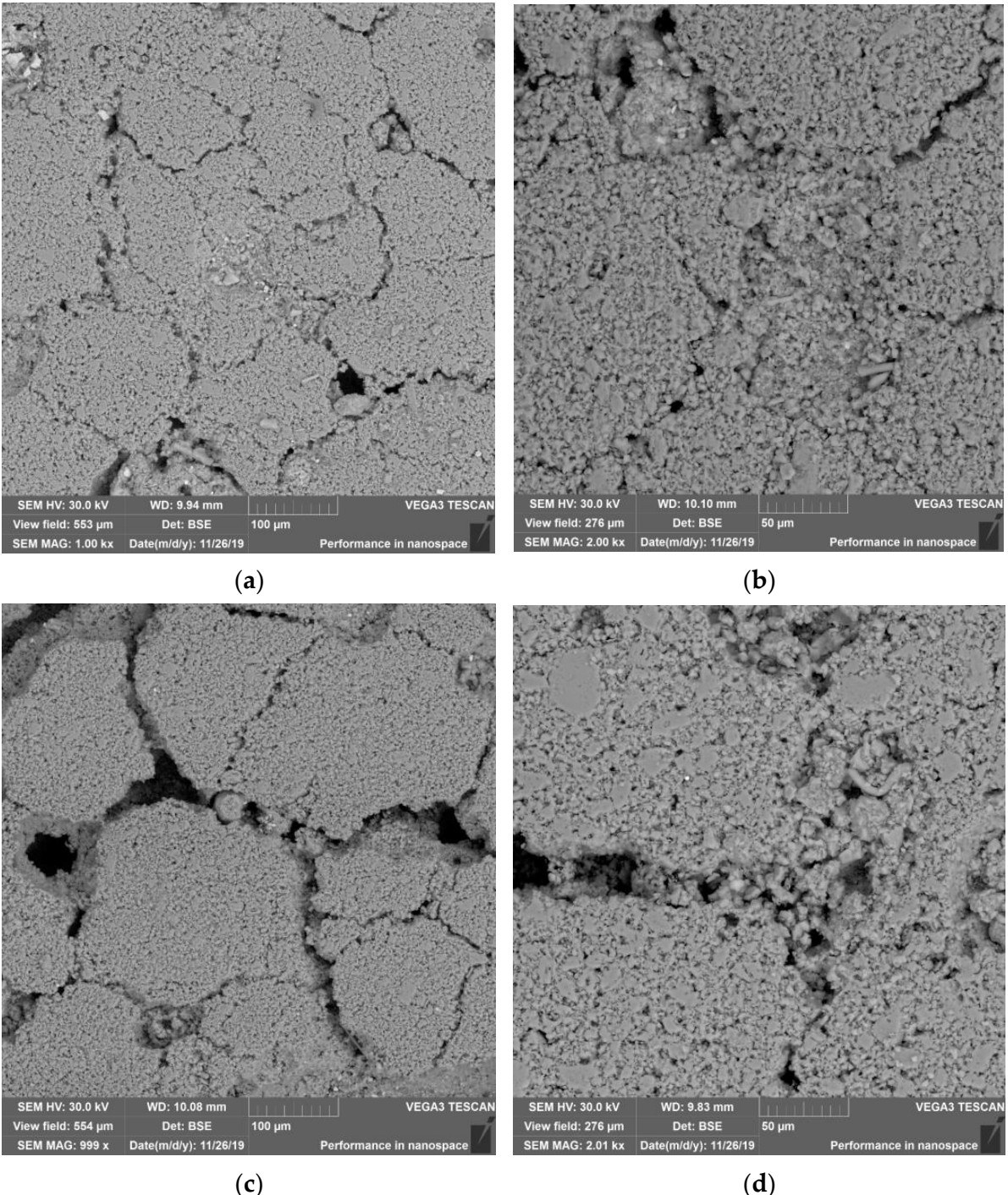

**Figure 22.** Ex situ SEM BSE images of accommodative compaction bands formed by densification of fragments in the interblock spaces of Sample #1 (**a**,**b**) and Sample #4 (**c**,**d**).

If the vector field is scanned along some contour and in each current position the circulation value is equal to Γ. Then, in the case when the contour completely falls into the vortex region, Γ = *const*.

In numerical form, Equation (5) can be written as follows [22]:

$$\Gamma_k = \oint_{L_k} (\vec{u}(x,y) \cdot \vec{dx}) \approx \sum_{i=0}^{0} \sum_{j=1}^{Ns-1} u_{x_j} \cdot \Delta x_j + \sum_{i=Ns}^{Ns} \sum_{j=1}^{Ns-1} u_{x_j} \cdot \Delta x_j + \sum_{j=0}^{0} \sum_{i=1}^{Ns-1} u_{y_i} \cdot \Delta y_i + \sum_{j=Ns}^{Ns} \sum_{i=1}^{Ns-1} u_{y_i} \cdot \Delta y_i \qquad (6)$$

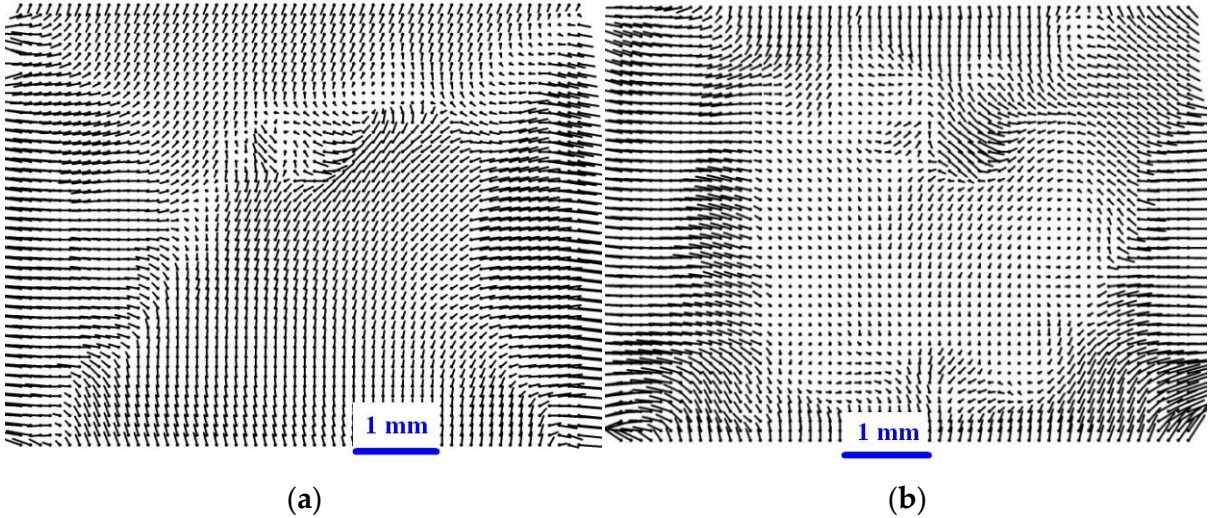

**Figure 23.** Fields of deformation vectors of segmented porous alumina: Sample #1, $\varepsilon = 0.077$, $\Delta\varepsilon = 0.010$ (**a**); Sample #4, $\varepsilon = 0.060$, $\Delta\varepsilon = 0.0063$ (**b**).

Here, $Ns$ is the length of the square-shaped contour. To simplify the calculations, the vectors falling on the corners of the squares will not be taken into account, and, therefore, given that $\Delta y_i = \Delta x_j = T$, Formula (6) can be rewritten [19]:

$$\Gamma_k = \left[\sum_{i=0}^{0}\sum_{j=1}^{Ns-1} u_{x_j} + \sum_{i=Ns}^{Ns}\sum_{j=1}^{Ns-1} u_{x_j} + \sum_{j=0}^{0}\sum_{i=1}^{Ns-1} u_{y_i} + \sum_{j=Ns}^{Ns}\sum_{i=1}^{Ns-1} u_{y_i}\right] T \tag{7}$$

When processing the experimental data, it should also be taken into account that each field vector is measured with a certain error. When calculating the circulation, the situation is possible when the number of vectors oriented in one direction (for example, counterclockwise) is commensurate with the number of vectors of the opposite orientation. To limit the influence of the error and increase the reliability of the calculation results, a parameter $\lambda$ is introduced in [22] that plays the role of a threshold. If this threshold is exceeded (the proportion of vectors in one of the directions is higher), then the circulation of this circuit is taken into account, and the program proceeds to the next contour.

In this work, this threshold was chosen as $\lambda = 1.05$, which means that the percentage of unidirectional vectors is exceeded by 5%. The calculation of the circulation was carried out at the spatial period of scanning by the contour $Td = 1$. Such a small period leads to the fact that all possible vortex motions are taken into account. On the other hand, this leads to the fact that, when moving to a new section, the program often takes into account previously calculated contours again. This approach makes it possible to compare different samples, but individual quantitative characteristics should be approached with caution.

The calculation of the total circulation of the field was performed on the basis of the seed point algorithm [22]. At each seed point, the characteristics of the vortex were calculated: circulation, its sign, and the size of the vortex. Then, the total circulation was found for each direction of rotation and its specific values (normalized to the number of vortex points). The direction of rotation was set by the variable sign.

Figure 24a,b show that evolution of the summary circulation with compression strain on Sample #1 can be represented by curve 1 with a maximum at $\varepsilon = 0.04$, i.e., at upgrowth portion, the stress–strain curve, where there is no anomaly, could explain this maximum. The strengthening factor vs. strain dependence (Figure 24b) demonstrates some small local minimum at $\varepsilon = 0.03$, which could be a forerunner of the circulation maximum at $\varepsilon = 0.04$.

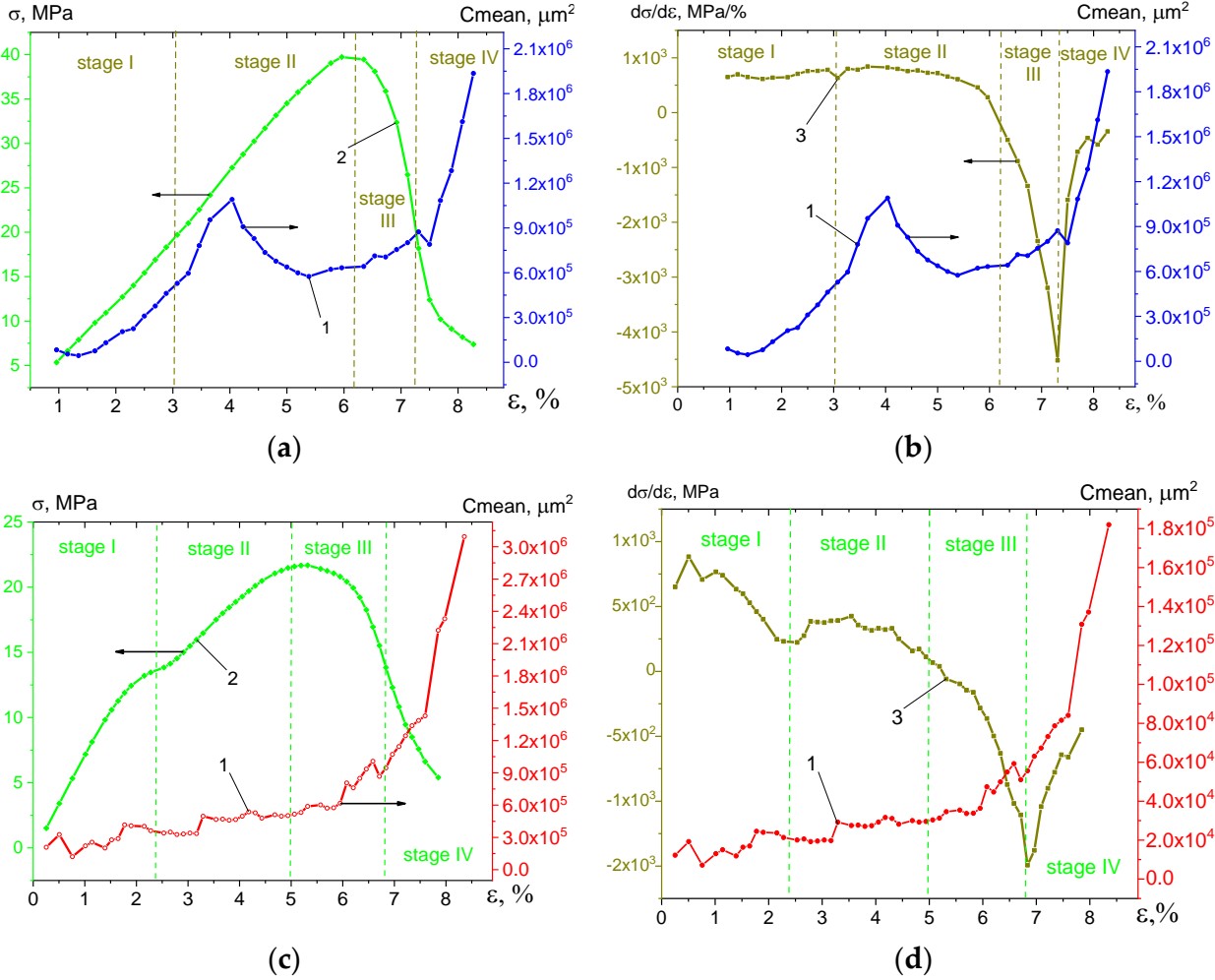

**Figure 24.** Strain-matched specific normalized circulation Cmean (1) with curves "*σ-ε*" (2) and "strain hardening- *ε*" (3) for Sample #1 (**a**,**b**) and for Sample #4 (**c**,**d**).

## 4. Discussion

### 4.1. Specifics of DIC Used in This Work

Traditionally, DIC is widely used for studying the deformation and fracture on various materials when the displacement field is obtained from processing two digital images taken from the material under loading at different moments of time. Frequently, the first image (reference) is obtained before the loading, while further ones (current) allow forming a series of vector fields. This approach allows the observation of integral strain accumulation in the region of interest, since the vector field is a result of time integration of the displacements. A ductile material may experience severe distortion during the loading, and, therefore, a new basic moment of time should be introduced with respect to which the damage accumulation and deformation would be examined. This approach calling for thoroughly selected time interval may be called a differential one.

It is common to use the integral DIC approach with the reference image of non-loaded sample for investigating deformation and fracture of ceramic materials characterized by pseudo-plasticity behavior [6,23–32]. In this work, we used subpixel resolution that allowed us to use the differential approach and obtain pseudo-images of deformation bands (Figures 5c–f, 6b–f, 8c–f, 9c–f, 11c–f and 12b–d). Let us note that there are no publications that would demonstrate the occurrence of the deformation bands on porous ceramic materials obtained with the use of the integral DIC approach.

Some authors, being engaged in studying deformation and fracture of porous ceramics, draw an analogy with deformation of porous rocks. It was reported [33] that deformation under the Hertz contact indentation loading of a high-porous (18%) brittle alumina ceramics is determined by structural breakdown, compaction, and intruding into the porous spaces with the following pore collapse and compaction banding similar to cataclastic flow formation of structures in rocks by propagation and coalescence of the inter-pore cracks, with the following filling the spaces by the material fragments.

The term "discrete compaction bands" was used [34] in discussing the loading curve with typical break-ups obtained on porous ceramic silicon dioxide foam. An analogy was drawn, then, with the behavior of limestone under loading.

In our work, the data were obtained about forming a whole set of shear bands whose appearance and strain localization degree depend on the bonding between segments (Figures 7, 10 and 13). The similar results on generation of a set of deformation bands have been described earlier in comparison between deformation of granulated media and particle fragmentation [35].

### 4.2. Low Damage Tolerance of Porous, Segmented Alumina

Highly porous ceramic materials are characterized by a three-stage behavior during compression [11–13]. At the first stage of compressive loading, these media show an almost linear behavior, which, however, allows the appearance of microcracks at the earliest stages of loading. Then follows the stage of inelastic deformation. At this stage, there are various micromechanisms of destruction of the spatial structure in which each pore is surrounded by thin walls of a dense material. Within this second stage, there is a gradual destruction of the walls between the pores and struts, and, therefore, the stress–strain curve shows some deviation from the linear behavior, sometimes even with the formation of a plateau [1–3]. Depending on the internal structure of the macrostructure, a highly porous material can withstand an increasing load with varying degrees of success, accumulating damage. Materials with high damage tolerance are usually able to preserve their spatial architecture due to the process of uniform microdestruction and microfragmentation of struts and walls between pores, which gradually captures a significant part of the internal volume of the sample [1–5]. The process of deformation of such a material is accompanied by a gradual increase in its density and Young's modulus due to the partial filling of the volume of the pore space with microfragments of the collapsed material of the walls and struts [2,36,37]. Thirdly Stage I is associated with the complete pore collapse and the beginning of the process of densification of the destroyed spatial structure.

The porous segmented ceramics studied in this work are characterized by a low damage threshold. Already, at the first stage, which ends at strain values corresponding to 0.5 UCS (for Sample #1) and 0.65UCS (for Sample #4), Figure 4a, a developed deformation structure consisting of from a family of compaction macrobands and accommodative compaction bands are formed in the samples (Figures 21 and 22).

Evolution of the $n = \varepsilon_{xx}/\varepsilon_{yy}$ ratio, with strain in Figure 20, can be additional evidence of low damage tolerance because the Poisson ratio value 0.9 at $\varepsilon = 1.5\%$ testifies that the elastic strain in Stage I is over, and the inelastic deformation stage has already started.

The presence of the segment boundary network around the microporous segments (Figure 2) offers ready crack propagation routes under compression. These cracks can easily grow by destructing the partial bonds between the segments formed by sintering. On the other hand, these segments are free to independently slide with respect to each other, as well as to experience microfragmentation of their boundaries under loading and thus dissipate the mechanical energy in friction without forming any trans-segment cracks. Such an inelastic damage tolerance behavior can be compared with that demonstrated by other porous ceramics and rocks that are usually interpreted in terms of crack opening/closing and compaction banding [38–47].

Given that the segmented porous ceramics revealed a non-linear tolerance behavior under loading due to accumulation of damage without the catastrophic fracture both before

Stage I and Stage II, Figure 4, and after Stage III and Stage IV, Figure 4, this results in achieving the maximum stress level, and this ceramic material may be classified as both damageable and a strain-softening ceramic.

*4.3. Segment Bonding Effect on Inelastic Deformation*

The coarse-grain alumina-base ceramic is a typical brittle material without any intrinsic toughening mechanism, nevertheless, it may be awarded with an extrinsic toughening mechanism, such as its internal spatial structure composed of segments and partially healed channels. The results are that such a toughening mechanism serves to provide fracture tolerance by letting the controlled displacement of the segments with increasing friction.

Summing up all the results, the evolution of the segmented porous alumina under compression loading can be described as follows. Strain localization serves for microcracking along the segment boundaries until forming primary macrocracks, whose orientation is determined by the triaxial stress–strain state. These macrocracks are observed as macrobands and form a dissipative structure that allows relative displacement between the segments with microcracking, fragmentation, and then compaction banding. The latter results in increasing friction between the fragments and thus arresting the crack propagation. Then, stress is redistributed, and new strain localization zones appear following the same path as above until the full sample's volume would be occupied by the compaction macrobands. Simultaneously, the accommodation compaction bands are formed around the segments and in large pores.

The preliminary characterization of porous segmented ceramics for strength and elasticity was reported [16], which demonstrated their high ability to accumulate damage and to show increased tolerance to existing and emerging defects, which makes it possible to effectively dissipate the mechanical energy due to formation of compaction bands during the compression. Ceramic Sample #1 and #4 clearly have this ability, with Sample #1 showing its great advantage over Sample #4 in that Sample #1's material segments are able to efficiently dissipate energy by collectively offsetting the segments relative to each other (Figures 17, 21 and 24a,b). At the same time, the inner structure of the segments is composed of ceramic grains and small pores, which also may experience pore collapse and fragmentation at some smaller scale levels, as compared to that of primary segments (Figures 2c,d and 22a,b).

It draws attention that deformation bands in Sample #1 are oriented at 45° with respect to the compression test axis (Figures 5e,f, 8b–d, 11b–d and 14b–d), whereas, in Sample #4, they are practically parallel to it (Figures 6e,f, 9d–e, 12b–d and 15b–d).

One can say that the improved bonding between segments in Sample #1, as compared to that of Sample #4, allow for demonstrating a higher degree of its structure involvement into inelastic deformation process with the progressive accumulation of microdamages throughout the entire volume of the sample. The partially healed segment boundaries provide higher strength in Sample #1 and allow redistribution of stress after primary crack formation, while in Sample #4, there are only weak bonds between the segment boundaries that cannot add up to the strength and accumulate damage very intensively. The partially healed channels in Sample #1 resist the cracking until some moment of time when primary macrocracks emerge, and this is the reason for CPS bump formation in Sample #1 at $\varepsilon = 4\%$ (Figure 17a). It seems that these high stress levels serve for launching the microcracking and fragmentation after primary crack formation and, namely, this activity is reflected by the maximum circulation values at this strain (Figure 24a,b). Reduction of both average CPS and circulation Cmean with strain may be provided by the following mechanical energy dissipation and stress redistribution due to intensive compaction banding (Figure 21) and crack retardation.

It was noted above that Sample #4 was not capable of such a redistribution and dissipation and, therefore, progressively accumulated damages (Figures 7b, 10b, 13b, 16b, 17, 19 and 24c,d), and the cardinal plastic shear dependence on strain is similar to that of

brittle alumina ceramics when shear magnitude is monotonically increased with the strain until complete fracture [48].

*4.4. Potential Applications*

Although mechanical strength is often a key parameter for many applications (e.g., implants, catalyst supports) where porous ceramic parts are subjected to compression, bending, and shear. There are few such studies in the literature where the porosity subsystem of a certain scale, which is a component of the hierarchical architecture, would not only play a functional role, but also improve the mechanical resistance of ceramics to emerging defects.

The ceramic materials obtained in this work, combining high strength and tolerance to defects, existing and formed under external influence, will expand the use of ceramics in a wide range of applications from filters and membranes to bone tissue endoprostheses, i.e., in those areas where it is necessary that the object has, in addition to the functionality associated with the presence of a large pore volume, also sufficient mechanical strength so that the ceramic component can be mechanically processed and integrated into the required engineering system with the porous material attached to other components using fittings and fasteners.

**5. Conclusions**

In this study, the 2D DIC technique has been applied to sequences of images taken from the surfaces of porous, segmented alumina samples during uniaxial compression tests. The sintered alumina was structurally composed of polycrystalline alumina grains with interior ~3–5-$\mu$m pores, as well as a network of discontinuities that subdivided the sample into ~230 $\mu$m segments and ~110-$\mu$m pores located at the discontinuity network nodes. Bimodal pore structure and the segment boundaries were the results of the evaporation and outgassing of the paraffin and ultra-high-molecular-weight polyethylene admixed with alumina powder via slip casting. Only partial bonding bridges between the segments were formed during a low-temperature sintering at 1300 °C for 1 h. A special technological approach made it possible to change the strength of the partial bonding bridges between the segments, which significantly affected the deformation behavior of porous, segmented aluminum during compression

One can see that DIC allows not only obtaining parameters, such as strain localization in in situ maps, cardinal plastic shear, and circulation of vector fields, but also reveals an extra key deformation stage related to structural toughening at the "$\sigma$-$\varepsilon$" curve at $\varepsilon \approx 4\%$. Along with compaction banding observed by the plastic shear curve, there is a maximum of the circulation characteristic, which serves to identify the vortex mode displacements that occur on the segment boundaries during microfragmentation and densification.

Therefore, the segmented ceramics are capable of effective stress redistribution under compression loading until full decomposition. Such a redistribution occurs as a relay effect of crack opening, microcracking, and microfragmentation, followed by compaction banding of both main cracks and accommodation ones on the segment boundaries.

In contrast to this, ceramics with weak bonding and wide segment boundaries do not have enough strength to resist decomposition by a stress redistribution mechanism, and, thus, they only continuously accumulate the damage.

**Author Contributions:** Conceptualization, V.K., N.S. and S.T.; methodology, V.K. and A.S.; software, V.K and A.S.; validation, V.K and A.S.; formal analysis, V.K., N.S. and S.T.; investigation, V.K., N.S., M.G., A.B. (Alexander Burlachenko) and A.B. (Ales Buyakov); resources, A.Z. and V.R.; data curation, V.K. and N.S.; writing—original draft preparation, V.K., N.S. and S.T.; writing—review and editing, V.K., N.S. and S.T.; visualization, V.K., N.S., M.G., A.B. (Alexander Burlachenko) and A.B. (Ales Buyakov); supervision, V.K., N.S. and S.T.; project administration, A.Z.; funding acquisition, A.Z. and V.R. All authors have read and agreed to the published version of the manuscript.

**Funding:** The work was performed according to a government research assignment for ISPMS SB RAS, projects FWRW-2021-0006 and FWRW-2021-0012.

**Institutional Review Board Statement:** Not applicable.

**Informed Consent Statement:** Not applicable.

**Data Availability Statement:** Not applicable.

**Acknowledgments:** The investigations have been carried out using the equipment of Share Use Centre "Nanotech" of the ISPMS SB RAS.

**Conflicts of Interest:** The authors declare no conflict of interest.

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
