# Peer review of "Digital Image Correlation Characterization of Deformation Behavior and Cracking of Porous Segmented Alumina under Uniaxial Compression"

_ceramics, doi:10.3390/ceramics6010008_

Round 1
Reviewer 1 Report
I am pleased to reviewed the manuscript that increase the advanced knowledge and deep studied in materials science. For my suggestion It will be great if you can apply this study with difference density of alumina sample. I have only two point would you to consider, 1 The annealing stage in line 94 should be replace with other word because annealing mean reduce stress from body. 2 Line 392 can you change the word from russian to english.
Author Response
1 The annealing stage in line 94 should be replace with other word because annealing mean reduce stress from body.
A: Corrected
2 Line 392 can you change the word from russian to english.
A: Corrected

Reviewer 2 Report
1. at page 13, line #392 : Russian ---> English
2. at page 26, line #611 : --- allow for for -----> allow for
Author Response
- at page 13, line #392 : Russian ---> English
A: Corrected
- at page 26, line #611 : --- allow for for -----> allow for
A: Corrected

Reviewer 3 Report
This research work discusses an interesting topic about a strategy to digital image correlation characterization of deformation behavior and cracking of porous segmented aluminа under uniaxial compression. This special technological approach significantly affects the deformation behavior of porous and segmented aluminum during compression. The background is explained well enough to follow the highlights of the work. The manuscript is appropriate for the journal if the authors comply with the suggestions listed below.
1. It is should be no more than five keywords in the manuscript, normally.
2. There is no the scale in some optical macrographs such as Figs. 6, 7 & 9 etc. please add them.
3. Why in this work the porous segmented ceramicsis was characterized by a low damage threshold, please make the corrsponding explanation.
Author Response
- It is should be no more than five keywords in the manuscript, normally.
A: Corrected
- There is no the scale in some optical macrographs such as Figs. 6, 7 & 9 etc. please add them.
A: Corrected
- Why in this work the porous segmented ceramicsis was characterized by a low damage threshold, please make the corresponding explanation.
A: This ceramic belongs to segmented materials with weak interfaces between segments. Already at low loads during compression, microfracture processes take place along the boundaries between the segments. This makes it possible to avoid strong localization of deformation in one place (and subsequent catastrophic failure), and contributes to the involvement of the entire volume of the material in the process of deformation and failure.

Reviewer 4 Report
The authors have studies the DIC studies on Alumina under uniaxial compression. The DIC study is the new features in the work and the authors studies the important property for ceramic materials. I request the authors to do the following changes.
1. The abstract can be consider to cut short and represent only the important findings in the work. Need not say about the experimental condition. Furthermore, the author must read the journal guidelines clearly to prepare the draught of the manuscripts.
2. Dont start the introduction with it is known that. It is known for few not for many. the language need to undergo full edit in the article.
3. It is shown and for example these sort of the text must be removed. Only 5 references cited in the intro. Cite the references sequentially.
4. Why did the authors selected the small qty of mixture in Injection molding. How did the authors control the wastage or minimize the process parameters to achieve the better the properties.
5. The assumptions of the eqn and the process need to be discussed in details. All the figure captions must be rewrite and the results must be correlated with the existing literatures.
6. The microstructure of the study is not clear enough to understand or appreciate the results and in what way the features can be understand or how th authors will distinguish the processing parameters?
7. the cross head speed and strain condition, sample dimension details are not listed out in the manuscript.
8. how many test samples were used? some of the image is mentioned with out scale bar and the quality of image looks dull with crack formation.?
9. There are plenty of figures with least discussions. the author must discuss and correlate with results.
Author Response
- The abstract can be consider to cut short and represent only the important findings in the work. Need not say about the experimental condition. Furthermore, the author must read the journal guidelines clearly to prepare the draught of the manuscripts.
A: Corrected
- Dont start the introduction with it is known that. It is known for few not for many. the language need to undergo full edit in the article.
A: Corrected
- It is shown and for example these sort of the text must be removed. Only 5 references cited in the intro. Cite the references sequentially
A: Corrected
- Why did the authors selected the small qty of mixture in Injection molding. How did the authors control the wastage or minimize the process parameters to achieve the better the properties.
A: We used commercial slip VK95-1 (Kontur, Cheboksary, Russia) for hot injection molding. As a rule, it is prepared with the introduction of 11.5-15 wt.% paraffin. Unfortunately, we did not find English information about this, can only offer a link to a Russian-language patent that describes the composition of such a slip. https://www.freepatent.ru/patents/2373169
- The assumptions of the eqn and the process need to be discussed in details. All the figure captions must be rewrite and the results must be correlated with the existing literatures.
A: Corrected
- The microstructure of the study is not clear enough to understand or appreciate the results and in what way the features can be understand or how th authors will distinguish the processing parameters?
A: Corrected
- the cross head speed and strain condition, sample dimension details are not listed out in the manuscript.
A: Cylindrical samples with a diameter of 10 mm and a height of 7 mm were subjected to mechanical compression tests at a loading speed of 2×10−4 s−1 (the cross-head speed of 0.1 mm/min). This is how the results for Figure 3a were obtained.
To carry out the DIC, a cylindrical surfaceÑ‹ was ground flat and polished to obtain a rectangular 7.2 mm × 5.8 mm flat field of interest (FOI). Such samples were also subjected to mechanical compression tests at a loading speed of 2×10−4 s−1 (the cross-head speed of 0.1 mm/min). This is how the results for Figure 4a were obtained.
- how many test samples were used? some of the image is mentioned with out scale bar and the quality of image looks dull with crack formation.?
A: Three to five sample test samples were used to obtain reliable data on the compressive mechanical properties with respect to the selected modes of obtaining segmented ceramics. In other words, three to five test samples were used to mechanically validate each of the listed heating modes (Sample#1-#4). This information has been added to Section 2.1. Sample Preparation and Examination. Scale bars added. The brightness-contrast ratio has been changed in the photographs.
- There are plenty of figures with least discussions. the author must discuss and correlate with results.
A: Corrected

Round 2
Reviewer 4 Report
The authors significantly improved the manuscript. However, the discussion section is lacking in the manuscript's current format. Although there are approximately 24 figures, not all of them have been discussed in detail or correlated with the available literatures. The authors must pay close attention to this detail.
Author Response
Reviewer 4
The authors significantly improved the manuscript. However, the discussion section is lacking in the manuscript's current format. Although there are approximately 24 figures, not all of them have been discussed in detail or correlated with the available literatures. The authors must pay close attention to this detail.
A: Added
